# FLD: Fourier Latent Dynamics for Structured Motion Representation and Learning

**Chenhao Li, Elijah Stanger-Jones, Steve Heim, Sangbae Kim**
Department of Mechanical Engineering, Massachusetts Institute of Technology
`{chenhli, elijahsj, sheim, sangbae}@mit.edu`

## Abstract

Motion trajectories offer reliable references for physics-based motion learning but suffer from sparsity, particularly in regions that lack sufficient data coverage. To address this challenge, we introduce a self-supervised, structured representation and generation method that extracts spatial-temporal relationships in periodic or quasi-periodic motions. The motion dynamics in a continuously parameterized latent space enable our method to enhance the interpolation and generalization capabilities of motion learning algorithms. The motion learning controller, informed by the motion parameterization, operates online tracking of a wide range of motions, including targets unseen during training. With a fallback mechanism, the controller dynamically adapts its tracking strategy and automatically resorts to safe action execution when a potentially risky target is proposed. By leveraging the identified spatial-temporal structure, our work opens new possibilities for future advancements in general motion representation and learning algorithms.

## 1 Introduction

The availability of reference trajectories, such as motion capture data, has significantly propelled the advancement of motion learning techniques (Peng et al., 2018; Bergamin et al., 2019; Peng et al., 2021; 2022; Starke et al., 2022; Li et al., 2023b;a). However, it is difficult to generalize policies using these techniques to motions outside the distribution of the available data (Peng et al., 2020; Li et al., 2023a). A core reason is that, while the trajectories in the data itself are induced by some dynamics of the system, the learned policies are typically trained to only replicate the data, instead of understanding the underlying dynamics structure. In other words, the policies attempt to memorize the trajectory instances rather than learn to *predict* them systematically. Consequently, the gaps between trajectories present challenges for these models in accurately representing and learning motion interpolations or transitions, resulting in limited generalizations (Wiley & Hahn, 1997; Rose et al., 1998). Moreover, the high nonlinearity and the embedded high-level similarity hinder data-driven methods from effectively identifying and modeling the dynamics of motion patterns (Peng et al., 2018). Therefore, addressing these challenges requires systematic understanding and leveraging the structured nature of the motion space.

Instead of handling raw motion trajectories in long-horizon, high-dimensional state space, structured representation methods introduce certain inductive biases during training and offer an efficient approach to managing complex movements (Min & Chai, 2012; Lee et al., 2021). These methods focus on extracting the essential features and temporal dependencies of motions, enabling more effective and compact representations (Lee et al., 2010; Levine et al., 2012). The ability to understand and capture the spatial-temporal structure of the motion space offers enhanced interpolation and generalization capabilities that can augment training datasets and improve the effectiveness of motion generation algorithms (Holden et al., 2017; Iscen et al., 2018; Ibarz et al., 2021). By uncovering and utilizing the underlying patterns and relationships within the motion space, continuous and rich sets of motions can be produced that progress realistically in a smooth and temporally coherent manner (Starke et al., 2020; 2022; 2023).

In this work, we present Fourier Latent Dynamics (FLD), a generative extension to Periodic Autoencoder (PAE) (Starke et al., 2022) that extracts spatial-temporal relationships in periodic or quasi-periodic motions with a novel predictive structure. FLD efficiently represents high-dimensional

trajectories by featuring motion dynamics in a continuously parameterized latent space that accommodates essential features and temporal dependencies of natural motions. The enforcement of latent dynamics empowers FLD to enhance the proficiency and generalization capabilities of motion learning algorithms with accurately described motion transitions and interpolations. The motion learning controllers, informed by the latent parameterization space of FLD, demonstrate extended online tracking capability. A novel fallback mechanism enables the learning agent to dynamically adapt its tracking strategy, automatically identifying and responding to potentially risky targets by rejecting and reverting to safe action execution. Finally, combined with adaptive learning algorithms, FLD presents strong long-term learning capabilities in open-ended learning tasks, strategically navigating and advancing through novel target motions while avoiding unlearnable regions.

In summary, our contributions include: **(i)** A self-supervised, structured representation and generation method featuring continuously parameterized latent dynamics for periodic or quasi-periodic motions. **(ii)** A motion learning and online tracking framework empowered by the latent dynamics with a fallback mechanism. **(iii)** Supplementary analysis of long-term learning capability with adaptive target sampling on open-ended motion learning tasks. Supplementary videos and more details for this work are available at `https://sites.google.com/view/iclr2024-fld/home`.

## 2 RELATED WORK

Motions are commonly described as long-horizon trajectories in high-dimensional state space. However, directly associating motions with raw trajectory instances yields highly inefficient representations and poor generalization that fail to capture motion features (Watter et al., 2015; Finn et al., 2016). In comparison, representing motions in a *structured* manner allows learning algorithms to better comprehend the underlying patterns and relationships within the motion space.

While a straightforward approach is to parameterize motions from physical dynamics, determining explicit models with correct dynamics structures solely from kinematic observations can be challenging without prior knowledge of the underlying physics (Li et al., 2023a). Structured trajectory generators address this issue by providing motion controllers with parameterized references with sufficient kinematic information. Some classical locomotion controllers parameterize the movement of each actuated degree of freedom by relying on cyclic open-loop trajectories described by sine curves (Tan et al., 2018) or central pattern generators (Ijspeert, 2008; Gay et al., 2013; Dörfler & Bullo, 2014; Shafiee et al., 2023). Recent works enable dynamical adaptation of high-level motion by having the control policy directly modulate the parameters of the trajectory generator, thus modifying the dictated trajectories (Iscen et al., 2018; Lee et al., 2020; Miki et al., 2022). In contrast to explicitly defined trajectory parameters, self-supervised models such as autoencoders explain motion evolution in a latent space. These representation methods have shown success in controlling non-linear dynamical systems (Watter et al., 2015), enabling complex decision-making (Ha & Schmidhuber, 2018), solving long-horizon tasks (Hafner et al., 2019), and imitating motion sequences (Berseth et al., 2019). A recent practice attempts to identify motion dynamics in a common latent space to foster temporal consistency between different dynamical systems (Kim et al., 2020).

Another line of structured motion representation involves extracting trajectory features in the frequency domain. Frequency domain methods have been proposed for various motion-related tasks, including synthesis (Liu et al., 1994), editing (Bruderlin & Williams, 1995), stylization (Unuma et al., 1995; Yumer & Mitra, 2016), and compression (Beaudoin et al., 2007). To consider the correlation between different body parts, a recent work on PAE constructs a latent space using an autoencoder structure and applies a frequency domain conversion as an inductive bias (Starke et al., 2022). The extracted latent parameters have been tested as effective full-body state representations in downstream motion learning tasks (Starke et al., 2023). Despite such progress, PAE is restricted to representing local frames and is not fully exploited to express overall motions or predict them.

## 3 PRELIMINARIES

Despite the success of self-supervised learning schemes in solving complex tasks, existing self-supervised learning methods inevitably overlook the intrinsic periodicity in data (Yang et al., 2022). Less attention has been paid to designing algorithms that capture prevalent periodic or quasi-periodic temporal dynamics in robotic tasks. To this end, we explore representation methods with an explicit

account of periodicity inspired by PAE (Starke et al., 2022) and develop generative capabilities thereon.

PAE addresses the challenges of learning the structure of the motion space, such as data sparsity and the highly nonlinear nature of the space, by focusing on the periodicity of motions in the frequency domain. The structure of PAE is illustrated in Fig. S7a. We denote trajectory segments of length $H$ in $d$-dimensional state space preceding time step $t$ by $\mathbf{s}_t = (s_{t-H+1}, \ldots, s_t) \in \mathbb{R}^{d \times H}$, as the input to PAE. The autoencoder structure decomposes the input motions into $c$ latent channels that accommodate lower-dimensional embedding $\mathbf{z}_t \in \mathbb{R}^{c \times H}$ of the motion input. A following differentiable Fast Fourier Transform obtains the frequency $f_t$, amplitude $a_t$, and offset $b_t$ vectors of the latent trajectories, while the phase vector $\phi_t$ is computed with a separate fully connected layer. We denote this parameterization process with $p$, we have

$$\mathbf{z}_t = \mathbf{enc}(\mathbf{s}_t), \quad \phi_t, f_t, a_t, b_t = p(\mathbf{z}_t), \tag{1}$$

where $\phi_t, f_t, a_t, b_t \in \mathbb{R}^c$. Next, the reconstructed latent trajectory segments $\hat{\mathbf{z}}_t \in \mathbb{R}^{c \times H}$ are computed using sinusoidal functions parameterized by the latent vectors with

$$\hat{\mathbf{z}}_t = \hat{p}(\phi_t, f_t, a_t, b_t) = a_t \sin\left(2\pi(f_t \mathcal{T} + \phi_t)\right) + b_t, \tag{2}$$

where $\hat{p}$ denotes the reconstruction, $\mathcal{T}$ is the time window corresponding to the state transition horizon $H$. Finally, the network decodes the reconstructed latent trajectories $\hat{\mathbf{z}}_t$ to the original motion space, and the reconstruction error is computed with respect to the original input

$$\hat{\mathbf{s}}_t = \mathbf{dec}(\hat{\mathbf{z}}_t), \quad L_0 = \text{MSE}(\hat{\mathbf{s}}_t, \mathbf{s}_t), \tag{3}$$

where $\hat{\mathbf{s}}_t \in \mathbb{R}^{d \times H}$, and MSE denotes the Mean Squared Error. The network structure of PAE is described in Table A.1.2. We refer to the original work (Starke et al., 2022) for more details.

PAE extracts a multi-dimensional latent space from full-body motion data, effectively clustering motions and creating a manifold in which computed feature distances provide a more meaningful similarity measure compared to the original motion space as visualized in Fig. 4.

## 4 APPROACH

### 4.1 PROBLEM FORMULATION

We consider the state space $\mathcal{S}$ and define a motion sequence $\tau = (s_0, s_1, \ldots)$ drawn from a reference dataset $\mathcal{M}$ as a trajectory of consecutive states $s \in \mathcal{S}$. Our research focuses on creating a physics-based learning controller capable of not only replicating motions prescribed by the reference dataset but also generating motions accordingly in response to novel target inputs, thereby enhancing its generality across a wide range of motions beyond the reference dataset. To this end, we adopt a two-stage training pipeline. In the first stage, an efficient representation model is trained on the reference dataset and a continuously parameterized latent space is obtained where novel motions can be synthesized by sampling the latent encodings. The second stage involves developing an effective learning algorithm that tracks the diverse generated target trajectories. In both motion representation and generation, we highlight the importance of identifying periodic or quasi-periodic changes in the underlying temporal progression of the motions that commonly describe robotic motor skills.

### 4.2 FOURIER LATENT DYNAMICS

By inspecting the parameters of the latent trajectories of periodic or quasi-periodic motions encoded by PAE, we observe that the frequency, amplitude, and offset vectors stay nearly time-invariant along the trajectories. We introduce the quasi-constant parameterization assumption.

**Assumption 1** *A latent trajectory* $\mathbf{z} = (\mathbf{z}_t, \mathbf{z}_{t+1}, \ldots)$ *can be approximated by* $\hat{\mathbf{z}} = (\hat{\mathbf{z}}_t, \hat{\mathbf{z}}_{t+1}, \ldots)$ *with a bounded error* $\delta = \|\mathbf{z} - \hat{\mathbf{z}}\|$, *where* $\hat{\mathbf{z}}_{t'} = \hat{p}(\phi_{t'}, f, a, b)$, $\forall t' \in \{t, t+1, \ldots\}$.

Assumption 1 holds with low approximation errors for periodic or quasi-periodic input motion trajectories, which yield constant frequency domain features. Since these latent features are learned, the assumption can be explicitly enforced. In the following context, we denote by $\phi_t$ the *latent state* and $f, a, b$ the *latent parameterization*. Here, we introduce Fourier Latent Dynamics (FLD), which

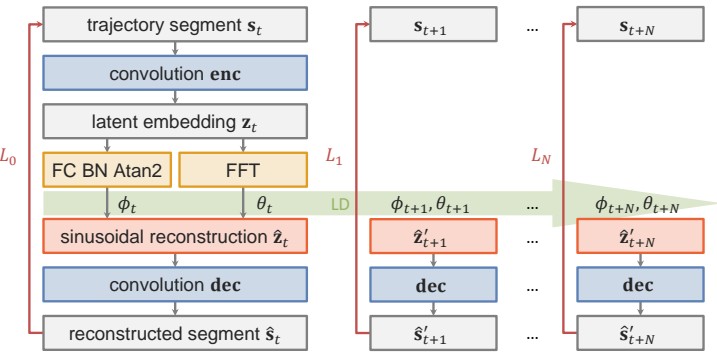

Figure 1: FLD training pipeline. During training, latent dynamics are enforced to predict proceeding latent states and parameterizations. The prediction loss is computed in the original motion space with respect to the ground truth future states.

enforces reconstruction of $\mathbf{z}$ over the complete trajectory by propagating latent dynamics parameterized by a local state $\phi_t$ and a *constant* set of global parameterization $f$, $a$, and $b$.

We formalize the latent dynamics of FLD and its training process in Fig. 1. For a motion segment $\mathbf{s}_t = (s_{t-H+1}, \ldots, s_t)$ whose latent trajectory segment $\mathbf{z}_t$ is parameterized by $\phi_t$, $f_t$, $a_t$, and $b_t$, we approximate the proceeding motion segment $\mathbf{s}_{t+i} = (s_{t-H+1+i}, \ldots, s_{t+i})$ with the prediction $\hat{\mathbf{s}}'_{t+i}$ decoded from $i$-step forward propagation $\hat{\mathbf{z}}'_{t+i}$ using the latent dynamics from time step $t$.

$$\hat{\mathbf{z}}'_{t+i} = \hat{p}(\phi_t + i f_t \Delta t, f_t, a_t, b_t), \quad \hat{\mathbf{s}}'_{t+i} = \mathbf{dec}(\hat{\mathbf{z}}'_{t+i}), \tag{4}$$

where $\Delta t$ denotes the step time. The latent dynamics in Eq. 4 assumes locally constant latent parameterizations and propagates latent states by advancing $i$ local phase increments. We can compute the prediction loss at time $t + i$. In fact, the local reconstruction process employed by PAE can be viewed as regression on a zero-step forward prediction using the latent dynamics. We can perform regressions on multi-step forward prediction by propagating the latent dynamics and define the total loss for training FLD with the maximum propagation horizon of $N$ and a decay factor $\alpha$,

$$L_{FLD}^N = \sum_{i=0}^{N} \alpha^i L_i, \quad L_i = \mathrm{MSE}(\hat{\mathbf{s}}'_{t+i}, \mathbf{s}_{t+i}). \tag{5}$$

Training with the FLD loss enforces Assump. 1 in a local range of $N$ steps. By choosing the appropriate maximum propagation horizon and the decay factor, one can balance the tradeoff between the accuracy of local reconstructions and the globalness of latent parameterizations. In comparison to PAE ($N = 0$), which lacks temporal propagation structure and performs only local reconstruction with local parameterization, FLD dramatically reduces the dimensions needed to express the entire trajectory, which facilitates both motion representation and generation. The latent dynamics also enable autoregressive motion synthesis. Starting from an initial latent state and parameterization, FLD generates future motion trajectories in a smooth and temporally coherent manner by propagating the latent dynamics and continually decoding the predicted latent encodings following Eq. 4.

For the following discussions, we consider training FLD on the reference dataset $\mathcal{M}$ and define the latent parameterization space $\Theta \subseteq \mathbb{R}^{3c}$ encompassing the latent frequency, amplitude, and offset. Therefore, each motion trajectory can be exclusively represented by a time-dependent latent state $\phi_t \in \mathbb{R}^c$ that describes the local time indexing and a constant latent parameterization $\theta = (f, a, b) \in \mathbb{R}^{3c}$ that describes the global high-level features of the motion. We establish this idea with a schematic view of the latent manifold induced by $\phi_t$ and $\theta$ in Suppl. A.1.3.

## 4.3 MOTION LEARNING

Given reference trajectories, physics-based motion learning algorithms train a control policy that actuates the joints of the simulated character or robot and reproduces the instructed motion trajectories. FLD is able to represent rich sets of motions efficiently. In contrast to discrete or handcrafted motion indicators, the feature distances in the continuously parameterized latent space of FLD provide learning algorithms a more reasonable similarity measure between motions.

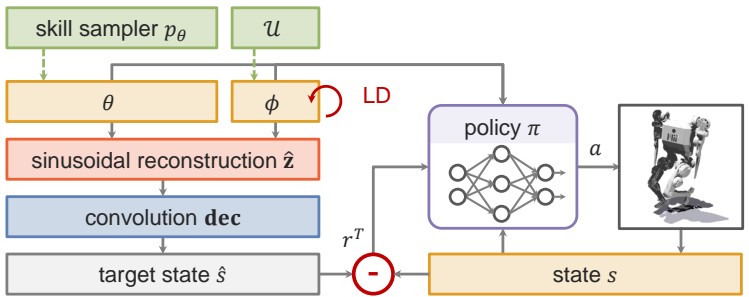

Figure 2: System overview. During training, the latent states propagate under the latent dynamics and are reconstructed to policy tracking targets $\hat{s}$ at each step. The tracking reward $r^T$ is computed as the distance between the target $\hat{s}$ and the measured states $s$.

### 4.3.1 POLICY TRAINING

At the beginning of each episode, a set of latent parameterization $\theta_0 \in \mathbb{R}^{3c}$ is sampled from a skill sampler $p_\theta$ (e.g. a buffer of offline reference motion encodings, more variants and ablation studies are detailed in Suppl. A.2.6). The latent state $\phi_0 \in \mathbb{R}^c$ is uniformly sampled from a fixed range $\mathcal{U}$. The step update of the latent vectors follows the latent dynamics in Eq. 4,

$$\theta_t = \theta_{t-1}, \quad \phi_t = \phi_{t-1} + f_{t-1}\Delta t. \tag{6}$$

At each step, the latent state and the latent parameterization are used to reconstruct a motion segment

$$\hat{\mathbf{s}}_t = (\hat{s}_{t-H+1}, \ldots, \hat{s}_t) = \mathbf{dec}(\hat{\mathbf{z}}_t) = \mathbf{dec}(\hat{p}(\phi_t, \theta_t)), \tag{7}$$

whose most recent state $\hat{s}_t$ serves as a tracking target for the learning environment at the current time step. The tracking reward encourages alignment with the target and is formulated in Suppl. A.2.7.

The latent state and parameterization are provided to the observation space to inform the policy about the motion and the specific frame it should be tracking. The policy observation and action space are described in Suppl. A.2.1. Figure 2 provides a schematic overview of the training pipeline, and an algorithm overview is detailed in Algorithm 1.

### 4.3.2 ONLINE TRACKING AND FALLBACK MECHANISM

During the inference phase, the policy structure incorporates real-time motion input as tracking targets, irrespective of their periodic or quasi-periodic nature. The latent parameterizations of the intended motion are obtained online using the FLD encoder. Figure S12 provides a schematic overview of the online tracking process. However, arbitrary reference inputs distant from the training distribution can result in limited tracking performance and potentially hazardous motor execution. Consequently, an online evaluation process is essential to assess the safety of a dictated motion, along with providing a fallback mechanism to ensure the availability of an alternative safe target when necessary. To address this requirement, FLD naturally induces a process that leverages the central role of latent dynamics. Consider the input sequence consisting of trajectory segments $\mathbf{s}_t^i = (s_{t-H+1}^i, \ldots, s_t^i)$ at each time step. These segments are stored in an input buffer $\mathcal{I}_t = (\mathbf{s}_{t-N+1}^i, \ldots, \mathbf{s}_t^i)$ of length $N$. Given the current latent state $\phi_t$, parameterization $\theta_t$, and the input buffer $\mathcal{I}_t$, our algorithm aims to determine the values of $\phi_{t+1}$ and $\theta_{t+1}$ for the subsequent step.

In the absence of user input, represented by $\mathcal{I}_t = \emptyset$, the latent parameters simply propagate the latent dynamics based on Eq. 4. Conversely, when user input is present, we perform reconstruction and forward prediction on the state segments stored in the input buffer $\mathcal{I}_t$, based upon the earliest recorded segment $\mathbf{s}_{t-N+1}^i$. The prediction is evaluated using the same loss metric $L_{FLD}^N$ employed in Eq. 5 to measure the dissimilarity between predicted and actual state trajectories within $\mathcal{I}_t$. This loss metric provides a quantification of the disparity in *dynamics* between motions, focusing on the similarity in state transitions and system evolution. Consequently, a small value of $L_{FLD}^N$ indicates that the input motion adheres to comparable spatial-temporal relationships and exhibits periodicity akin to those observed in the training dataset.

To determine the suitability of an input tracking target for acceptance or rejection, we establish a threshold $\epsilon_{FLD}$ derived from training statistics. When an input motion is accepted, the updated

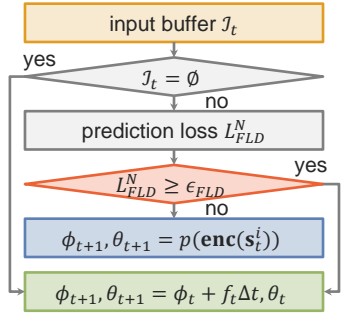
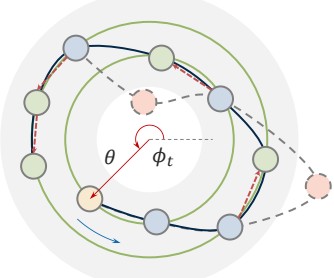

(a) Target evaluation process  (b) Schematic overview

Figure 3: Online tracking and fallback mechanism. (a) The prediction loss $L_{FLD}^N$ is evaluated within an input buffer of user-proposed tracking targets. The mechanism accepts the proposal only when the prediction loss is below a threshold $\epsilon_{FLD}$. (b) The proposed tracking targets (dashed curve) may contain risky states (dashed red dots). The fallback mechanism identifies these states and defaults them to safe alternatives (dashed red arrows and green dots) by propagating latent dynamics. Note that the real-time tracking trajectories are not necessarily periodic or quasi-periodic.

tracking target is encoded from the latest state trajectory segment $\mathbf{s}_t^i$ within the input buffer $\mathcal{I}_t$. This evaluation process is formulated in Fig. 3a. Additionally, Fig. 3b presents a schematic overview of the online tracking and fallback mechanism.

## 5 EXPERIMENTS

We evaluate FLD on the MIT Humanoid robot (Chignoli et al., 2021), with which we show its applicability to state-of-the-art real-world robotic systems. We use the human locomotion clips collected in Peng et al. (2018) retargeted to the joint space of our robot as the reference motion dataset containing slow and fast jog, forward and backward run, slow and fast step in place, left and right turn, and forward stride. We visualize some representative motions in Fig. S9. Note that the motion labels are not observed during the training of the models and are only used for evaluation. In the motion learning experiments, we use Proximal Policy Optimization (PPO) (Schulman et al., 2017) in Isaac Gym (Rudin et al., 2022). Suppl. A.1 and Suppl. A.2 provide further training details.

### 5.1 STRUCTURED MOTION REPRESENTATION

We compare the motion embeddings of the reference dataset obtained from training FLD following Sec. 4.2 with different models, with parameters detailed in Suppl. A.1.2. The state space is specified in Suppl. A.1.1. After the computation of the latent manifold, we project the principal components of the phase features onto a two-dimensional plane, as outlined in (Starke et al., 2022). We then compare the latent structure induced by FLD with that by PAE. Additionally, we adopt a Variational Autoencoder (VAE) as a commonly employed method for representing motions in a lower-dimensional space. Lastly, we plot the principal components of the original motion states for comprehensive analysis. We illustrate the latent embeddings acquired by these models in Fig. 4, where each point corresponds to a latent representation of a trajectory segment input.

We elucidate the results by highlighting the inductive biases imposed during the training of these models. In the provided figures, samples from the same motion categories are assigned the same color, indicating a close relationship between neighboring frames within the embedding. This *spatial* relationship is implicitly enforced by the reconstruction process of all the models, promoting latent encodings of close frames to remain proximate in the latent space. However, the degree of *temporal* structure enforcement varies significantly, owing to the differing inductive biases.

Notably, FLD demonstrates the most consistent structure akin to concentric cycles, primarily due to the motion-predictive structure within the latent dynamics enforced by Eq. 5. The cycles depicted in the figures represent the primary period of individual motions. The angle around the center (latent state) signifies the timing, while the distance from the center (latent parameterization) represents the

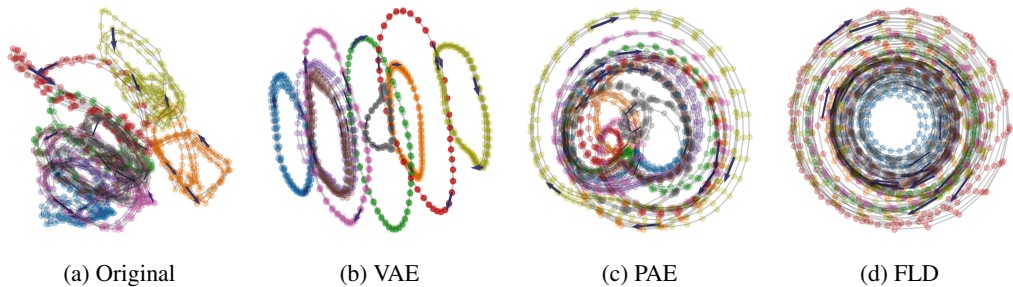

| (a) Original | (b) VAE | (c) PAE | (d) FLD |
|:---:|:---:|:---:|:---:|

Figure 4: Latent manifolds for different motions. Each color is associated with a trajectory from a motion type. The arrows denote the state evolution direction. FLD presents the strongest spatial-temporal relationships with explicit latent dynamics enforcement. PAE witnesses a similar but weaker pattern with local sinusoidal reconstruction. In comparison, VAE enables only spatial close-ness, and the trajectories of the original states are the least structured.

high-level features (e.g. velocity, direction, contact frequency, etc.) that remain consistent through-out the trajectory. This pattern reflects the strong temporal regularity captured by Assump. 1, which preserves time-invariant global information regarding the overall motion. As PAE can be viewed as FLD with zero-step latent propagation, in contrast, we observe a weaker pattern in the latent manifold of PAE, where the consistency of high-level features holds only locally. Finally, the re-construction process employed in VAE training does not impose any specific constraints on the temporal structure of system propagation. Consequently, the resulting latent representation, except for the direct encoding, exhibits the least structured characteristics among the models.

Powered by the latent dynamics, FLD offers a compact representation of high-dimensional motions by employing the time index vector $\phi_t$ and assuming high-level feature consistency $\theta$ throughout each trajectory. Conversely, PAE encodes motion features only locally $\theta_t = \theta(\phi_t)$. The numbers of parameters of different models used to express a trajectory of length $|\tau|$ are listed in Table 1.

Table 1: Motion representation parameters

| Model | Original | VAE | PAE | FLD |
|---|---|---|---|---|
| # Parameters | $d \times |\tau|$ | $c \times (|\tau| - H + 1)$ | $4c \times (|\tau| - H + 1)$ | $4c$ |

## 5.2 MOTION RECONSTRUCTION AND PREDICTION

We demonstrate the generality of FLD in reconstructing and predicting *unseen* motions during train-ing. Figure 5 (left) illustrates a representative validation with a diagonal run motion. At time $t = 65$, FLD undertakes motion reconstruction and prediction for future state transitions based on the most recent information $\mathbf{s}_t$, as elaborated in Sec. 4.2. For comparison, we train a PAE and a feed-forward (FF) model with fully connected layers with the same input and output structure as FLD.

It is evident that the motion predicted by FLD aligns with the actual trajectories. Particularly in joint position evolution which presents strong sinusoidal periodicity, it exhibits the lowest relative error $e$. The superiority of FLD is especially pronounced in long-term prediction regions, where the other models accumulate significantly larger compounding errors. The effectiveness of FLD in accurately predicting motion for an extended horizon is attributed to the latent dynamics enforced with an appropriate propagation horizon $N$ in Eq. 5. In the extreme case of $N = 0$ (PAE), the relative error is larger due to the weaker temporal propagation structure. The result on the diagonal run trajectory demonstrates the ability of FLD to accurately predict future states despite not being exposed to this specific motion during training. This showcases the generalization capability of FLD, as it effectively captures the underlying dynamics and temporal relationships inherent in the training dataset, which are prevalent and can be adapted to unseen motions. In comparison, the FF model training fails to understand the spatial-temporal structure in the motions and results in strong

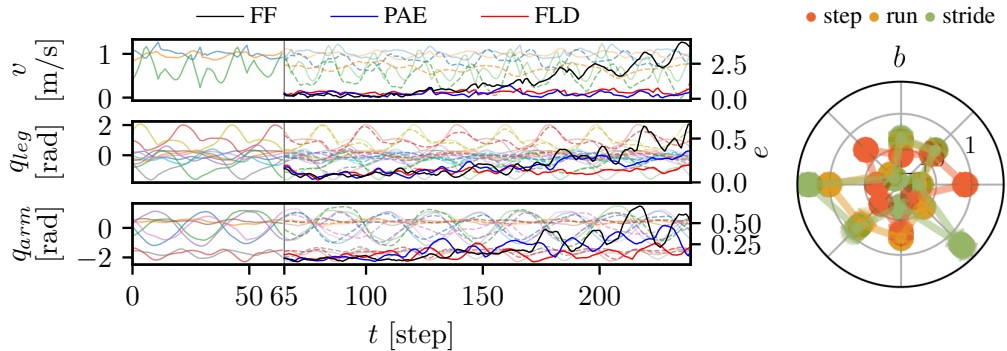

Figure 5: Motion reconstruction and prediction of a diagonal run trajectory (left). The solid and dashed curves denote the ground truth and predicted state evolution. The relative prediction error (vivid) of FF, PAE and FLD is depicted with the axis indicating $e$ on the right. Latent offset (right) of step in place, forward run, and forward stride. Each radius denotes a latent channel.

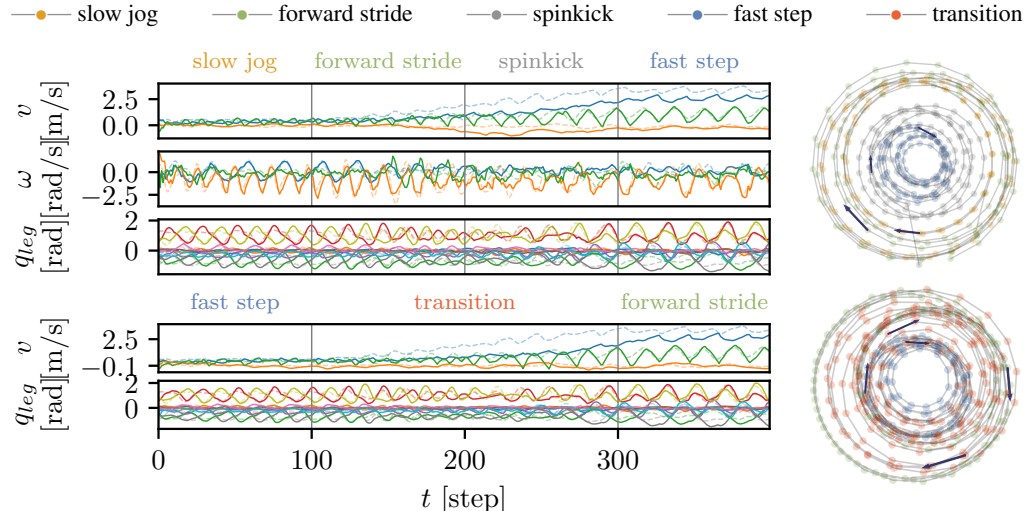

Figure 6: Motion tracking and fallback (top) and motion transition (bottom). The dashed curves denote the user-specified tracking target, and the solid ones denote the measured system states. The corresponding latent manifolds are depicted on the right side.

overfitting to the training dataset, thus limiting its generality. Moreover, the dedicated FF model solely propagates the states through autoregression and does not provide any data representation.

With the embedded motion-predictive structure, the enhanced generality achieved by FLD is attributed to the well-shaped latent representation space, where sensible distances between motion patterns are established. Figure 5 (right) depicts the latent offsets of step in place, forward run, and forward stride, where the parameterization of the intermediate motion (forward run) is distributed in between. This high-level understanding of motion similarity is further exemplified in Fig. S10 on different motion types (illustrated in Fig. S9) and forward run velocities.

## 5.3 MOTION TRACKING AND FALLBACK

Following Sec. 4.3, we can learn a motion tracking controller that employs FLD parameterization space. We perform an online tracking experiment where real-time user input of various motion types is provided to the controller as tracking targets.

In the first example (Fig. 6, top), we switch the input motion to a different type every 100 time steps (indicated by vertical grey lines). Notably, one of the input motions, referred to as spinkick (visualized in Fig. S17c), lives far from the training distribution and is considered a risky input. We

observe that the controller achieves accurate user input tracking, as evidenced by the close alignment between the dictated (dashed) and measured (solid) states, except for the spinkick motion. At time $t = 200$, when the proposed states of the spinkick motion are received, FLD evaluates the latent dynamics loss $L_{FLD}^N$ and rejects the proposal. This decision is based on the limited similarity calculated between the proposed system evolution and the state propagation prevalent in the training dataset. In response to the rejected motion, the fallback mechanism is triggered, providing safe alternative reference states that extend from the previous motion. Consequently, the controller continues to track the forward stride motion, with the actual reference states indicated by the dashed curves from the previous region.

Moreover, the controller demonstrates the ability to transition between different tracking targets smoothly. By considering the tracking of an arbitrary motion as a process of wandering between continuously parameterized periodic priors, FLD dynamically extracts essential characteristics of local approximates. To further understand the performance of FLD and the learning agent on tracking motions that fall into the gaps between trajectories captured in the reference dataset, we construct in the second example (Fig. 6, bottom) a transition phase where the target motion parameterizations are obtained from linear interpolation between the source and target motions. In particular, the interpolated movements exhibit a gradual evolution of high-level motion features, providing a clear and structured transition from high-frequency, low-velocity stepping to low-frequency, high-velocity striding sequences. This gradual evolution of motion features in the interpolated trajectories suggests that FLD is capable of capturing and preserving the essential temporal and spatial relationships of the underlying motions. It bridges the gap between different motion types and velocities, generating coherent and natural motion sequences that smoothly transition from one to another.

### 5.4 EXTENDED DISCUSSION: SKILL SAMPLER DESIGN

Our experiment provides strong evidence that the motion learning policy informed by the FLD latent parameterization space effectively achieves motion in-betweening and coherent transitions that encapsulate high-level behavior migrations. While this policy already shows remarkable performance when trained using targets derived from offline reference motions, its potential can be further amplified with skill samplers that can continually propose novel training targets, which lead to enhanced tracking generality onto a wider range of motions.

To this end, we perform ablation studies with different skill sampler implementations, including a learning-progress-based online curriculum (**ALPGMM**), and evaluate policy tracking generality beyond the reference dataset. The policy trained with **ALPGMM** demonstrates the ability to acquire knowledge of the continuously parameterized latent space through interactions with the environment and achieves enhanced performance in general motion tracking tasks as opposed to the fixed offline target sampling scheme (**OFFLINE**). We direct interested readers to Suppl. A.2.10 and Suppl. A.2.11 for supplementary experiments regarding this extended discussion.

## 6 CONCLUSION

In this work, we present FLD, a novel self-supervised, structured representation and generation method that extracts spatial-temporal relationships in periodic or quasi-periodic motions. FLD efficiently represents high-dimensional trajectories by featuring motion dynamics in a continuously parameterized latent space that accommodates essential features and temporal dependencies of natural motions. Compared with models without explicitly enforced temporal structures, FLD significantly reduces the number of parameters required to express non-linear trajectories and generalizes accurate state transition prediction to unseen motions. The enhanced generality by FLD is further confirmed with the high-level understanding of motion similarity by the latent parameterization space. The motion learning controllers, informed by the latent parameterization space, demonstrate extended online tracking capability. Our proposed fallback mechanism equips learning agents with the ability to dynamically adapt their tracking strategies, automatically recognizing and responding to potentially risky targets. Finally, our supplementary experiments on skill samplers and adaptive learning schemes reveal the long-term learning capabilities of FLD, enabling learning agents to strategically advance novel target motions while avoiding unlearnable regions. By leveraging the identified spatial-temporal structure, FLD opens up possibilities for future advancements in motion representation and learning algorithms.

REPRODUCIBILITY STATEMENT

The experiment results presented in this work can be reproduced with the dataset and implementation details provided in Suppl. A.1 and Suppl. A.2. The code has been open-sourced on the project page.

ETHICS STATEMENT

Our research on Fourier Latent Dynamics (FLD) involves constructing a meaningful parameterization to encode, compare, and predict spatial-temporal relationships in data inputs. While we focus on its applicability in robot motion learning tasks, we acknowledge the existence of concerns about the potential misuse of the model to a broader range of data types.

From our experience, this approach is primarily constrained by the availability of rich training data, along with accurate input data during run-time. Our reliance on data from motion-capture techniques, which necessitate a lab environment or simulations with direct ground-truth access, limits its practicality outside lab settings. However, this may change in the future with advancements in motion reconstruction from standard video feeds.

The model could also be used to generate plausible but fictitious actions for individuals, potentially leading to misinterpretations or misjudgments about their capabilities or intentions. If integrated into automated systems, FLD's predictive capabilities could influence decision-making processes. While this might enhance efficiency, it could pose risks if the model's predictions are inaccurate or based on biased data, leading to harmful decisions.

ACKNOWLEDGMENTS

We thank the members of the Biomimetic Robotics Lab for the helpful discussions and feedback on the paper. We are grateful to MIT SuperCloud and Lincoln Laboratory Supercomputing Center for providing HPC resources. This research was funded by NAVER Labs and Advanced Robotics Lab of LG Electronics Co., Ltd.

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

# A APPENDIX

## A.1 MOTION REPRESENTATION DETAILS

### A.1.1 STATE SPACE

The state space is composed of base linear and angular velocities $v$, $\omega$ in the robot frame, measurement of the gravity vector in the robot frame $g$, and joint positions $q$ as in Table S2.

Table S2: Policy observation space

| Entry | Symbol | Dimensions |
|-------|--------|------------|
| base linear velocity | $v$ | 0:3 |
| base angular velocity | $\omega$ | 3:6 |
| projected gravity | $g$ | 6:9 |
| joint positions | $q$ | 9:27 |

### A.1.2 REPRESENTATION TRAINING PARAMETERS

The learning networks and algorithm are implemented in PyTorch 1.10 with CUDA 12.0. Adam is used as the optimizer for training the representation models. The information is summarized in Table S3.

Table S3: Representation training parameters

| Parameter | Symbol | Value |
|-----------|--------|-------|
| step time seconds | $\Delta t$ | 0.02 |
| max iterations | — | 5000 |
| learning rate | — | 0.0001 |
| weight decay | — | 0.0005 |
| learning epochs | — | 5 |
| mini-batches | — | 10 |
| latent channels | $c$ | 8 |
| trajectory segment length | $H$ | 51 |
| FLD propagation horizon | $N$ | 50 |
| propagation decay | $\alpha$ | 1.0 |
| approximate training hours | — | 1 |

Periodic Autoencoder

FLD shares the same network architecture as PAE, whose encoder and decoder are composed of 1D convolutional layers. The periodicity in the latent trajectories is enforced by parameterizing each latent curve as a sinusoidal function. While the latent frequency, amplitude, and offset are computed with a differentiable real Fast Fourier Transform layer, the latent phase is determined using a linear layer followed by Atan2 applied on 2D signed phase shifts on each channel. The network architecture is detailed in Table S4.

Variational Autoencoder

VAE is implemented as our baseline for representing motions in a lower-dimensional space. In our comparison, it admits the same input, output data structure and the same latent dimension as PAE. The network architecture is detailed in Table S5.

Other baselines

The dedicated feed-forward model shares the same input and prediction output structure as FLD, however, it does not provide any representation of the motion it predicts. It is trained to evaluate the motion synthesis performance of FLD.

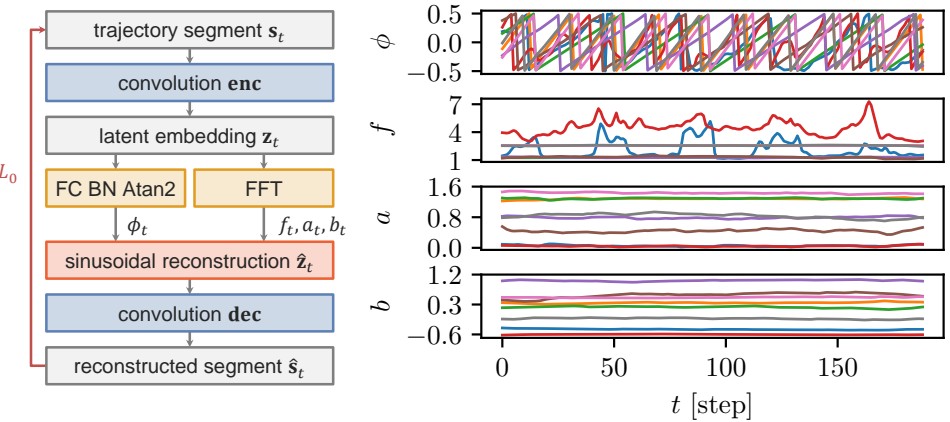

(a) PAE structure.                    (b) Latent parameters along a forward run trajectory.

Figure S7: Motion representation using PAE. (a) PAE utilizes frequency domain analysis to extract the local periodicity of highly nonlinear motions. Latent features are constructed using sinusoidal functions. (b) Each color is associated with a distinct latent channel. Despite the fluctuation on two frequency channels, the latent frequency $f$, amplitude $a$, and offset $b$ stay nearly constant throughout the trajectory.

Table S4: PAE architecture

| Network | Layer | Output size | Kernel size | Normalization | Activation |
|---|---|---|---|---|---|
| encoder | Conv1d | $64 \times 51$ | 51 | BN | ELU |
| | Conv1d | $64 \times 51$ | 51 | BN | ELU |
| | Conv1d | $8 \times 51$ | 51 | BN | ELU |
| phase encoder | Linear | $8 \times 2$ | — | BN | Atan2 |
| decoder | Conv1d | $64 \times 51$ | 51 | BN | ELU |
| | Conv1d | $64 \times 51$ | 51 | BN | ELU |
| | Conv1d | $27 \times 51$ | 51 | BN | ELU |

The oracle classifier is trained to predict original motion classes from their latent parameterizations for evaluation purposes. It is trained to provide a better understanding of the adaptive curriculum learning migration in the latent parameterization space using privileged motion type information.

The network architectures of these models are presented in Table S6.

### A.1.3  LATENT MANIFOLD

Each motion within the set $\mathcal{M}$ maintains a consistent latent parameterization $\theta$ throughout its trajectory. Consequently, its latent representation can be visualized as a circle (solid grey curves) in Fig. S8a, where $\theta$ represents the distance from the center. The current state at a specific time frame is denoted by the latent state $\phi_t$, which is depicted as the angle elapsed along the circle. The propagation of latent dynamics and motion evolution can thus be described as traveling around the circle. All motions within $\mathcal{M}$ correspond to a collection of circles that span a latent subspace, represented by the shaded rings. Therefore, we can interpolate or synthesize new motions by sampling within or around this latent subspace. It is important to note that this latent subspace may be ill-defined for policy learning, as it may contain subregions where the decoded motions describe challenging or unlearnable movements for the real learning system.

Table S5: VAE architecture

| Network | Type | Hidden | Activation |
|---|---|---|---|
| encoder | MLP | $512, 256, 128$ | ReLU |
| mean encoder | Linear | – | – |
| standard deviation encoder | Linear | – | Softplus |
| decoder | MLP | $128, 256, 512$ | ReLU |

Table S6: Other baseline architectures

| Network | Symbol | Type | Hidden | Activation |
|---|---|---|---|---|
| feed-forward | – | MLP | $512, 512$ | ELU |
| classifier | – | MLP | $1024, 512$ | ELU |

### A.1.4 REFERENCE MOTIONS

### A.1.5 SIMILARITY EVALUATION

Both figures clearly demonstrate a well-understood similarity between high-level motion features. In Fig. S10a, the latent representation of the intermediate motion (run, orange curves) is positioned between their neighboring representations (step in place and forward stride, red and green curves). A similar pattern is also identified in Fig. S10b, with a more consistent latent representation within the same motion type (run forward), compared to a more considerable structural discrepancy between different motion types in Fig. S10a. As a result of this well-understood spatial-temporal structure, sampling in the latent representation space leads to realistic transitions and interpolations. This ability to fill in the gaps and generate a rich set of natural motions from sparse offline datasets showcases the efficacy of FLD in motion generation tasks. In summary, the automatically induced latent representation space $\Theta$ in FLD plays a crucial role in capturing the underlying spatial-temporal relationships and facilitating realistic motion generation through interpolation and transition. This advancement enables the algorithm to learn from sparse offline data and generalize proficiently, greatly enriching available training datasets that improve the efficiency of the downstream learning process and the capability of learned policies.

### A.2 MOTION LEARNING DETAILS

### A.2.1 OBSERVATION AND ACTION SPACE

Besides the state space, the policy observes additional information such as joint velocities $\dot{q}$ and last action $a'$ in its observation space. Most importantly, the policy observes the instructed target motions by admitting their latent states and parameterizations. The detailed information together with the artificial noise added during training to increase the policy robustness is presented in Table S7.

The action space is of 18 dimensions and encodes the target joint position for each of the 18 actuators. The PD gains are set to $30.0$ and $5.0$, respectively.

### A.2.2 POLICY TRAINING PARAMETERS

Adam is used as the optimizer for the policy and value function with an adaptive learning rate with a KL divergence target of $0.01$. The policy runs at $50\,\mathrm{Hz}$. All training is done by collecting experiences from $4096$ uncorrelated instances of the simulator in parallel. The information is summarized in Table S8.

### A.2.3 NETWORK ARCHITECTURE

The network structures of the learning policy $\pi$ and the value function $V$ used in PPO training are detailed in Table S9.

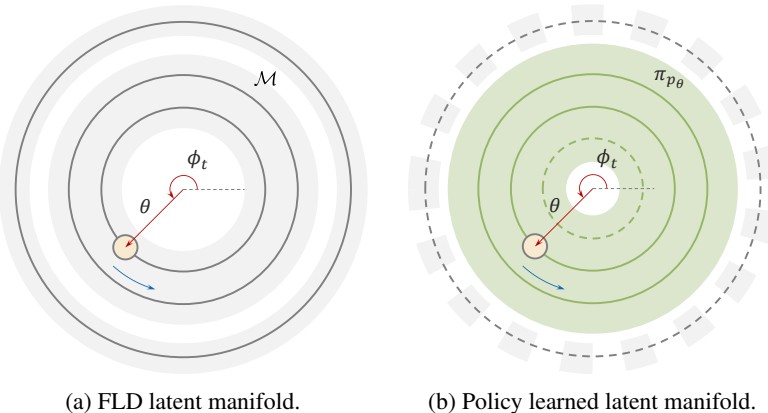

(a) FLD latent manifold.      (b) Policy learned latent manifold.

Figure S8: Schematic view of the latent manifold induced by the latent state and latent parameterization of FLD. While the latent parameterization $\theta$ determines which motion the current state is experiencing, the latent state $\phi_t$ indicates the time index of the state frame on this motion. (a) Each motion is represented by a solid grey circle. The shaded rings denote the collection of representations of motions in the offline dataset $\mathcal{M}$. (b) The grey circle denotes an unlearnable motion and the grey shaded ring denotes the unlearnable subspace. The green circles represent learnable motions, with the dashed one denoting a motion outside the offline dataset $\mathcal{M}$ but acquired during training. The green shaded ring denotes the motion region the policy eventually masters.

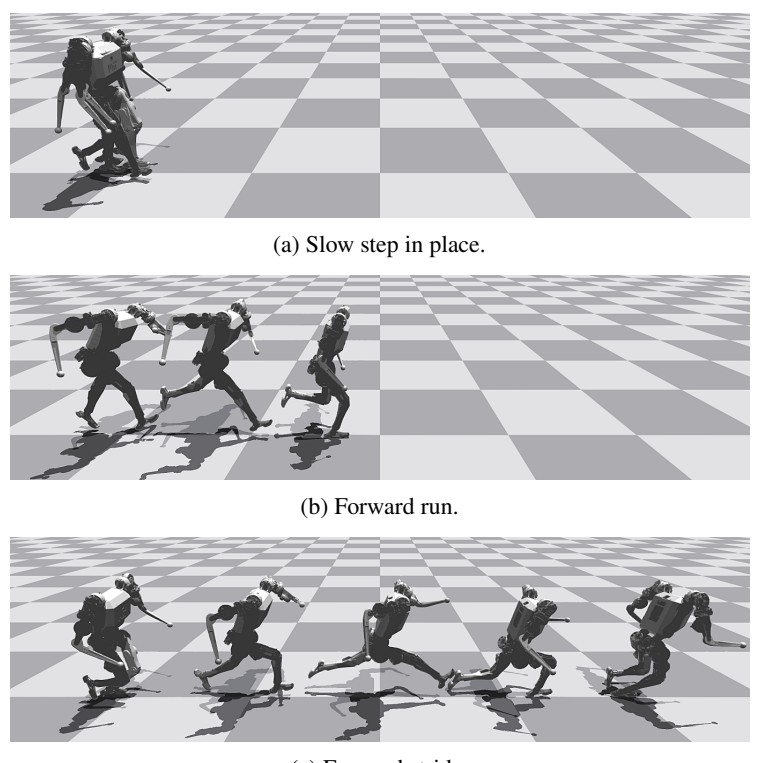

(a) Slow step in place.

(b) Forward run.

(c) Forward stride.

Figure S9: Representative motions in the training dataset. Base linear and angular displacement is integrated from velocity information.

### A.2.4 DOMAIN RANDOMIZATION

In addition to observation noise, domain randomization is applied during training to improve policy robustness for real-world application scenarios. On the one hand, the base mass of the parallel

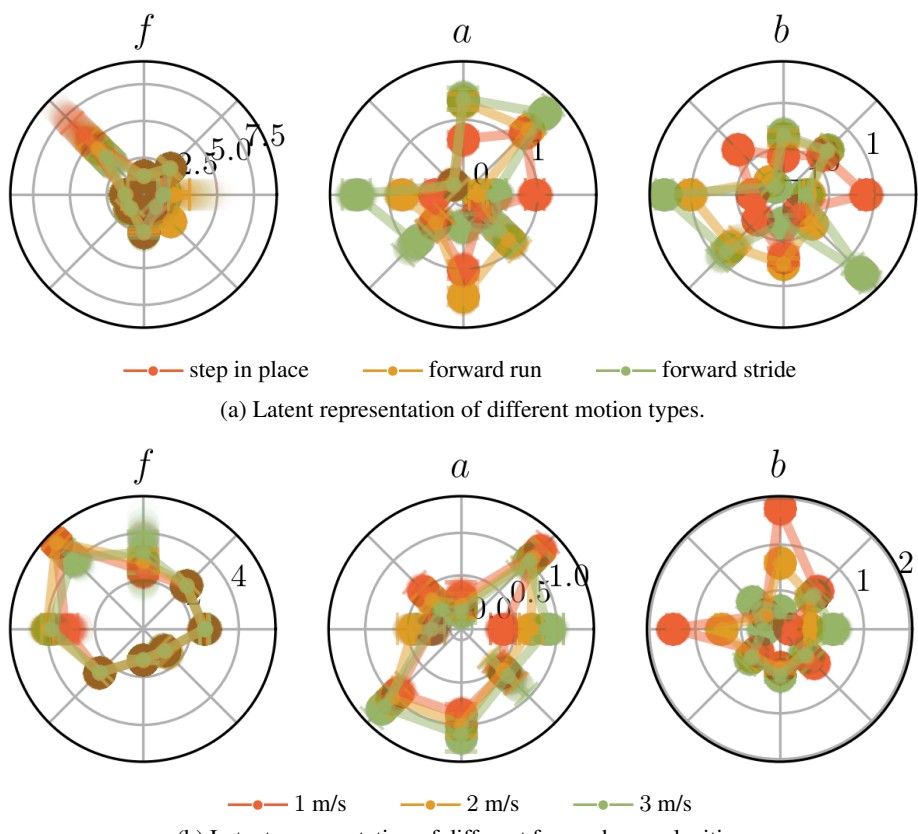

(a) Latent representation of different motion types.

(b) Latent representation of different forward run velocities.

Figure S10: Similarity evaluation and motion interpolation in the latent representation space. The connected lines across the latent channels indicate the mean latent representation of trajectories from the same motion or same velocity. The relative locations between representations of related trajectories illustrate the well-understood similarity between high-level motion features.

Table S7: Policy observation space

| Entry | Symbol | Dimensions | Noise level |
|-------|--------|------------|-------------|
| base linear velocity | $v$ | 0:3 | 0.2 |
| base angular velocity | $\omega$ | 3:6 | 0.05 |
| projected gravity | $g$ | 6:9 | 0.05 |
| joint positions | $q$ | 9:27 | 0.01 |
| joint velocities | $\dot{q}$ | 27:45 | 0.75 |
| last actions | $a'$ | 45:63 | 0.0 |
| latent phase | $\sin\phi$ | 63:71 | 0.0 |
| latent phase | $\cos\phi$ | 71:79 | 0.0 |
| latent frequency | $f$ | 79:87 | 0.0 |
| latent amplitude | $a$ | 87:95 | 0.0 |
| latent offset | $b$ | 95:103 | 0.0 |

training instances is perturbed with an additional weight $m' \sim \mathcal{U}(-0.5, 1.0)$, where $\mathcal{U}$ denotes uniform distribution. On the other hand, random pushing is also applied every 15 seconds on the robot base by forcing its horizontal linear velocity to be set randomly within $v_{xy} \sim \mathcal{U}(-0.5, 0.5)$.

### A.2.5 ALGORITHM OVERVIEW

Table S8: Policy training parameters

| Parameter | Symbol | Value |
|---|---|---|
| step time seconds | $\Delta t$ | 0.02 |
| skill-performance buffer size | $|\mathcal{B}|$ | 5000 |
| max episode time seconds | − | 20 |
| max iterations | − | 20000 |
| learning rate | − | 0.001 |
| steps per iteration | − | 24 |
| learning epochs | − | 5 |
| mini-batches | − | 4 |
| KL divergence target | − | 0.01 |
| discount factor | $\gamma$ | 0.99 |
| clip range | $\epsilon$ | 0.2 |
| entropy coefficient | − | 0.01 |
| parallel training environments | − | 4096 |
| number of seeds | − | 5 |
| approximate training hours | − | 2 |

Table S9: Policy training architecture

| Network | Symbol | Type | Hidden | Activation |
|---|---|---|---|---|
| policy | $\pi$ | MLP | $128, 128, 128$ | ELU |
| value function | $V$ | MLP | $128, 128, 128$ | ELU |

Algorithm 1 provides details of policy training. More information on the skill sampler and the skill-performance buffer $\mathcal{B}$ is given in Suppl. A.2.6.

### A.2.6 SKILL SAMPLERS

To achieve high tracking performance for various target motions, the design of the skill sampler $p_\theta$ is critical. Our sampling strategy, which guides policy learning and enhances tracking, is shown in Fig. S8b. Training FLD on the offline dataset $\mathcal{M}$ reveals latent parameterization spaces with unlearnable subspaces (grey dashed shaded ring), posing policy learning challenges. An effective skill sampler interacts with the environment during training to identify and avoid these subspaces, and it should also explore novel motion targets, expanding performance boundaries. Figure S8b shows the desired performance region (green shaded ring) covering many learnable motions (green solid circles) in $\mathcal{M}$ and extending to new, effectively learned motions (green dashed circle).

We compared policy learning performance using four skill samplers: offline point sampler (**OFFLINE**), offline Gaussian mixture model (GMM) sampler (**GMM**), random sampler (**RANDOM**), and absolute learning progress with GMM sampler (**ALPGMM**) (Portelas et al., 2020). **OFFLINE** and **GMM** access the original offline dataset $\mathcal{M}$, while **RANDOM** and **ALPGMM** directly interact and navigate the latent space $\Theta$ during online training. Their implementation details are presented in Suppl. A.2.6.

Offline point sampler

We can utilize the offline dataset to encode trajectories into points $\theta$ in the latent space $\Theta$, adding them to the buffer $p_\theta$. During online training, random latent parameterization points are drawn from this buffer, and the offline trajectories are recovered through the generation process of FLD. However, this sampler is limited by dataset size and memory constraints.

Offline Gaussian Mixture Model sampler

This method uses a GMM (Rasmussen, 1999) to parameterize the latent space $\Theta$, avoiding storing all points $\theta$. An offline GMM $p_\theta$ is fitted using the Expectation-Maximization (EM) (Dempster et al.,

---

**Algorithm 1** Policy training

---

1: **Input**: FLD decoder **dec**
2: initialize skill sampler $p_\theta$, skill-performance buffer $\mathcal{B}$
3: **for** learning iterations $= 1, 2, \ldots$ **do**
4:     sample latent states $\phi$ and latent parameterizations $\theta$
5:     **for** environment steps $= 1, 2, \ldots$ **do**
6:         generate motion tracking targets $\hat{s}$ from $\phi$ and $\theta$ with FLD decoder **dec**
7:         calculate tracking reward $r^T$
8:         propagate latent states $\phi$ according to Eq. 6
9:         **if** reset **then**
10:             collect latent parameterizations $\theta$ and the tracking performance $r_e^T$ in skill-performance buffer $\mathcal{B}$
11:             resample latent states $\phi$ and latent parameterizations $\theta$
12:         **end if**
13:     **end for**
14:     update skill sampler $p_\theta$ with skill-performance buffer $\mathcal{B}$
15:     update policy and value function with PPO or another RL algorithm
16: **end for**

---

1977) algorithm, with random points drawn during online training. The sampler's effectiveness depends on the dataset's quality, as it may struggle with unlearnable or challenging motions.

Random sampler

Without access to the original dataset, the random sampler draws samples uniformly from the latent space confidence region. However, it overlooks the data structure and may generate unlearnable targets due to inefficient understanding of large sensorimotor spaces.

Absolute Learning Progress with Gaussian Mixture Models sampler

Identifying learnable motions in the latent space without the original dataset is challenging. A self-adaptive curriculum learning approach, essential in this context, adjusts the frequency of sampling target motions based on their potential for improved learning. This method parallels the *strategic student problem* (Lopes & Oudeyer, 2012), focusing on selecting tasks for maximal competence. We employ the ALP-GMM strategy for this purpose. ALP-GMM fits a GMM on previously sampled latent parameterizations, linked to their ALP values. Sampling decisions are made using a non-stochastic multi-armed bandit approach (Auer et al., 2002), with each Gaussian distribution as an arm and ALP as its utility, steering the sampling towards high-ALP areas.

To get this per-parameterization ALP value $alp(\theta)$, we follow the implementation from earlier work (Portelas et al., 2020). For each newly sampled latent parameterization $\theta$ and associated tracking performance $r_{e,new}^T(\theta)$, the closest previously sampled latent parameterization with associated tracking performance $r_{e,old}^T(\theta)$ is retrieved from a skill-performance buffer $\mathcal{B}$ using the nearest neighbor algorithm. We then have

$$alp(\theta) = \mid r_{e,new}^T(\theta) - r_{e,old}^T(\theta) \mid . \tag{S8}$$

We use Faiss (Johnson et al., 2019) for efficient vector similarity search to find the nearest neighbors of the current latent parameterization in the skill-performance buffer $\mathcal{B}$. At each policy learning iteration, we update the skill sampler by refitting the online GMM based on the collected latent parameterizations and the corresponding ALP measure. We refer to the original work (Portelas et al., 2020) for more implementation details.

The design details of different skill samplers are presented in Table S10. For **ALPGMM**, we provide in Fig. S11 a schematic overview of its working pipeline.

Table S10: Skill sampler parameters

| Skill sampler | Parameter | Value |
|---|---|---|
| **OFFLINE** | features | 24 |
| | buffer size | 20000 |
| **GMM** | features | 24 |
| | components | 8 |
| | covariance type | full |
| **RANDOM** | features | 24 |
| **ALPGMM** | features | 25 |
| | minimum components | 2 |
| | maximum components | 10 |
| | information criterion | Bayesian |
| | random sample rate | 0.2 |
| | skill-performance buffer size | 5000 |
| | skill-performance collection interval | 5 |
| | nearest neighbors | 1 |
| | update interval | 50 |

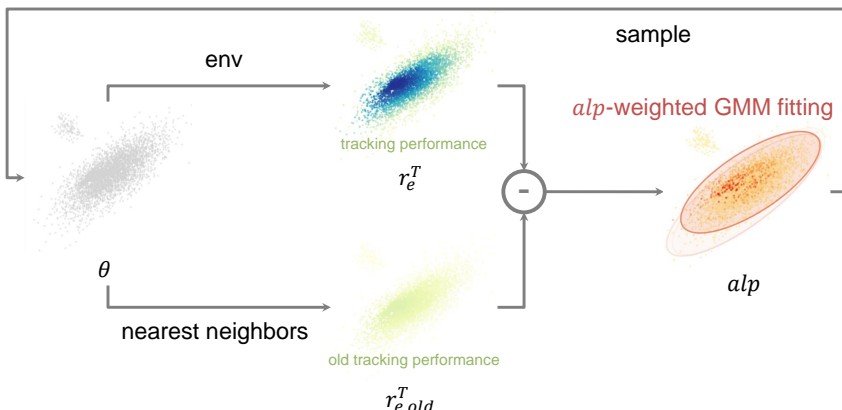

Figure S11: **ALPGMM** algorithm overview. The ALP is calculated on the latent parameterization of target motion samples by comparing the old and new tracking performance. A GMM is then fitted on these samples weighted by their ALP measure. New samples are drawn from the fitted GMM in the next iteration.

### A.2.7 TRACKING PERFORMANCE

The tracking reward calculates the weighted sum of reward terms on each dimension bounded in $[0, 1]$,

$$r^T = w_v r_v + w_\omega r_\omega + w_g r_g + w_{q_{leg}} r_{q_{leg}} + w_{q_{arm}} r_{q_{arm}}. \tag{S9}$$

The tracking performance $r_e^T \in [0, 1]$ is computed by the normalized episodic tracking reward. The formulation of each tracking term is detailed as follows and their weights in Table S11.

Table S11: Tracking reward weights

| Weight | $w_v$ | $w_\omega$ | $w_g$ | $w_{q_{leg}}$ | $w_{q_{arm}}$ |
|---|---|---|---|---|---|
| Value | 1.0 | 1.0 | 1.0 | 1.0 | 1.0 |

Linear velocity

$$r_v = e^{-\sigma_v \|\hat{v} - v\|_2^2}, \tag{S10}$$

where $\sigma_v = 0.2$ denotes a temperature factor, $\hat{v}$ and $v$ denote the reconstructed target and current linear velocity.

Angular velocity

$$r_\omega = w_\omega e^{-\sigma_\omega \|\hat{\omega} - \omega\|_2^2}, \tag{S11}$$

where $\sigma_\omega = 0.2$ denotes a temperature factor, $\hat{\omega}$ and $\omega$ denote the reconstructed target and current angular velocity.

Projected gravity

$$r_g = e^{-\sigma_g \|\hat{g} - g\|_2^2}, \tag{S12}$$

where $\sigma_g = 1.0$ denotes a temperature factor, $\hat{g}$ and $g$ denote the reconstructed target and current projected gravity.

Leg position

$$r_{q_{leg}} = e^{-\sigma_{q_{leg}} \|\hat{q}_{leg} - q_{leg}\|_2^2}, \tag{S13}$$

where $\sigma_{q_{leg}} = 1.0$ denotes a temperature factor, $\hat{q}_{leg}$ and $q_{leg}$ denote the reconstructed target and current leg position.

Arm position

$$r_{q_{arm}} = e^{-\sigma_{q_{arm}} \|\hat{q}_{arm} - q_{arm}\|_2^2}, \tag{S14}$$

where $\sigma_{q_{arm}} = 1.0$ denotes a temperature factor, $\hat{q}_{arm}$ and $q_{arm}$ denote the reconstructed target and current arm position.

### A.2.8 REGULARIZATION REWARDS

The regularization reward calculates the weighted sum of individual regularization reward terms.,

$$r^R = w_{ar} r_{ar} + w_{q_a} r_{q_a} + w_{q_T} r_{q_T}. \tag{S15}$$

The formulation of each regularization term is detailed as follows and their weights in Table S12.

Table S12: Regularization reward weights

| Weight | $w_{ar}$ | $w_{q_a}$ | $w_{q_T}$ |
|--------|----------|-----------|-----------|
| Value | $-0.01$ | $-2.5 \times 10^{-7}$ | $-1.0 \times 10^{-5}$ |

Action rate

$$r_{ar} = \|a' - a\|_2^2, \tag{S16}$$

where $a'$ and $a$ denote the previous and current actions.

Joint acceleration

$$r_{q_a} = \left\| \frac{\dot{q}' - \dot{q}}{\Delta t} \right\|_2^2, \tag{S17}$$

where $\dot{q}'$ and $\dot{q}$ denote the previous and current joint velocity, $\Delta t$ denotes the step time interval.

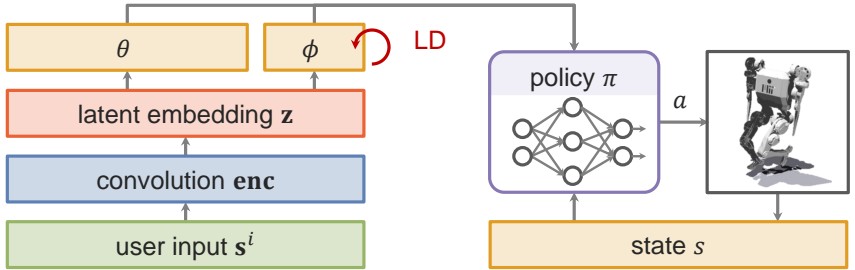

Figure S12: Online tracking. The latent states $\phi$ and parameterization $\theta$ are obtained from encoding accepted target proposals from user input $\mathbf{s}_t^i$. In cases where the user proposal is absent or rejected, the latent dynamics step in and continuously provide fallback targets.

Joint torque

$$r_{q_T} = \|T\|_2^2, \tag{S18}$$

where $T$ denotes the joint torques.

### A.2.9 ONLINE TRACKING

Figure S12 provides a schematic overview of the online motion tracking pipeline.

### A.2.10 ADAPTIVE CURRICULUM LEARNING

In addition to solely tracking targets in the offline dataset, we aim to address the challenge of learning continuous interpolation and transitions between motions in the sparsely populated reference motion space.

To this end, we train FLD on the offline reference dataset $\mathcal{M}$ in the first stage and obtain the latent parameterization space $\Theta$. In the second stage of online motion learning, we generate tracking targets by sampling and decoding latent parameterizations with a skill sampler $p_\theta$,

$$\theta = (f, a, b) \sim p_\theta \in \Delta(\Theta). \tag{S19}$$

Depending on the design of the skill sampler $p_\theta$, the training datasets with FLD motion synthesis may describe different spaces of dictated reference motions used to train the policy. The resulting policy trained over this dataset induced by $p_\theta$ is denoted by $\pi_{p_\theta}$.

We distinguish the performance evaluation spectrums on closed-ended and open-ended learning. We define *expert* performance as the performance evaluation of a learned tracking policy on the *prescribed* target motions. And we define *general* performance as the performance evaluation of a learned tracking policy on a *wide spectrum* of target motions.

Therefore, the objective of motion learning can be written as

$$\max_{\pi_{p_\theta}, p_\theta} \mathbb{E}_{\theta' \in \Theta'} \left[ V(\pi_{p_\theta} \mid \theta') \right], \tag{S20}$$

where $V(\pi_{p_\theta} \mid \theta')$ denotes the performance metric of policy $\pi_{p_\theta}$ on motion $\theta'$, and $\Theta'$ denotes the evaluation spectrum that differs in expert and general performances.

For expert and general tracking performances, we define the motion learning evaluation spectrum $\Theta'$ in Eq. S20. To evaluate the *expert* performance, we randomly sample test target motions from the collected latent parameterizations of the prescribed offline dataset $\mathcal{M}$. For the *general* performance, the evaluation is determined by uniformly random samples from the continuous latent parameterization space bounded by the confidence region induced by FLD training on $\mathcal{M}$. The result on five random seeds is presented in Table S13.

The **OFFLINE** sampler, with direct access to the dataset $\mathcal{M}$, generates trajectories closely resembling the target motion space, leading to high expert performance. However, it does not use FLD's capability to synthesize motions for broader tracking targets, limiting its generality to the policy

Table S13: Tracking performance with proposed skill samplers.

| Range | OFFLINE | GMM | RANDOM | ALPGMM |
|---|---|---|---|---|
| Expert | $0.88 \pm 0.03$ | $0.87 \pm 0.03$ | $0.83 \pm 0.01$ | $0.87 \pm 0.04$ |
| General | $0.72 \pm 0.01$ | $0.73 \pm 0.02$ | $0.80 \pm 0.01$ | $0.80 \pm 0.03$ |

Figure S13: Tracking performance and exploration factor on running motion targets.

network's understanding of spatial-temporal motion structures. The **GMM** sampler, similarly, uses parameterized latent representations of prescribed motions, achieving performance comparable to **OFFLINE**.

In contrast, the **RANDOM** sampler optimizes the policy for a wide range of random motions without specific targeting, resulting in a less focused approach. **ALPGMM**, however, balances generalization with performance on prescribed targets. It does this by learning from policy-environment interactions and identifying useful target motions during training. The ALP measure guides its GMM sampling strategy, avoiding redundant targets and focusing on promising, underexplored motions for significant performance improvements.

To understand this adaptive learning process, we maintain an online buffer $\tilde{\Theta}$ of the latent parameterization of running target motions selected by the skill samplers and evaluate the policies' average tracking performance on these motions. To quantify the online target search space enabled by the skill samplers, we define an exploration factor $\gamma(\tilde{\Theta})$ as a channel-averaged ratio between the standard deviation within the latent parameterizations of the online running target motion $\sigma_{\tilde{\Theta}}$ and that within the initial offline dataset.

$$\gamma(\tilde{\Theta}) = \mathbb{E}_c \left[ \frac{\sigma_{\tilde{\Theta}}}{\sigma_{\Theta}} \right]_c. \tag{S21}$$

We plot the results in Fig. S13 with the running tracking performance $r_e^T(\tilde{\Theta})$ in five random runs.

Note that the target motion space induced by different skill samplers and thus the spectrum the running tracking performance is calculated over is different. **OFFLINE** and **GMM**, with access to offline data, show strong tracking (bold curves) of prescribed motions but fail to explore beyond this dataset, keeping their exploration factors (thin curves) constant. In contrast, **RANDOM**, despite lacking offline data and understanding of data structure, achieves similar performance due to

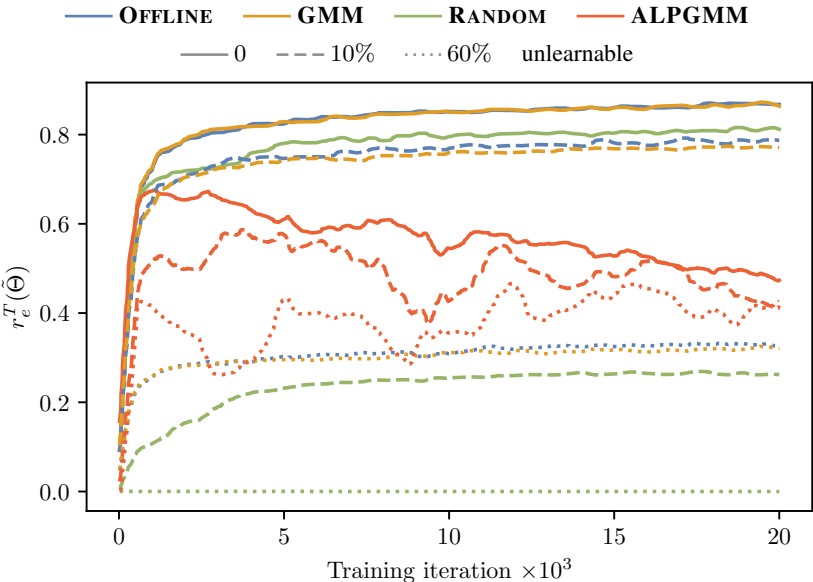

Figure S14: Tracking performance on running motion targets under different mixtures of unlearnable reference motions. The solid, dashed and dotted curves denote training with 0, 10% and 60% unlearnable motions in the reference dataset, respectively.

FLD's well-structured latent space. However, this strategy may fail if the initial training data has unlearnable components, as further discussed in Suppl. A.2.11.

**ALPGMM** initially shows improved tracking performance, which gradually decreases as it explores new motion regions, including under-explored or undefined areas. This expansion, evidenced by the increasing exploration factor, results in a much larger motion coverage by training's end. This guided exploration ensures maintained expert performance on mastered motions in Table S13, which would have been lost if the sampling region is grown only blindly. In our experiments, we keep the curriculum learning open-ended. Future studies may look into practical regularization methods to constrain such exploration to convergence. In summary, **ALPGMM** achieves high expert and general performance by dynamically adapting its sampling distribution during training, resulting in broader coverage of the target motion space and improved motion learning generality.

### A.2.11  UNLEARNABLE SUBSPACES

Similar to the discussion above, we investigate the average tracking performance over the running targets sampled by the skill samplers under datasets containing different mixtures of unlearnable reference motions. To this end, FLD and the latent parameterization space are pre-trained on each dataset in the first stage. We plot the mean of five random policy training runs on each dataset of 0, 10% and 60% unlearnable motions in Fig. S14, respectively.

The solid curves indicate a similar tracking performance evolution pattern as discussed above, where the latent parameterization space is not corrupted by the unlearnable motions. However, when the reference dataset contains unlearnable motions, blind random sampling strategies like **RANDOM** quickly fail. The presence of unlearnable motions distorts the latent parameterization space towards these challenging areas. Sampling strategies that rely heavily on direct access to the offline dataset (**OFFLINE**, **GMM**) or the ill-shaped space induced by it (**RANDOM**) may lead to a substantial number of unreachable targets and significant learning difficulties. Especially with 60% motions unlearnable, policies trained with **OFFLINE** and **GMM** achieve less than half of the tracking performance in the fully learnable case, and those with **RANDOM** encounter drastic learning failure.

In contrast, **ALPGMM** facilitates policy learning by actively adapting its sampling distribution towards regions with a high ALP measure, identifying promising targets and avoiding those with limited potential for improvement. This adaptive curriculum is particularly effective in cases where

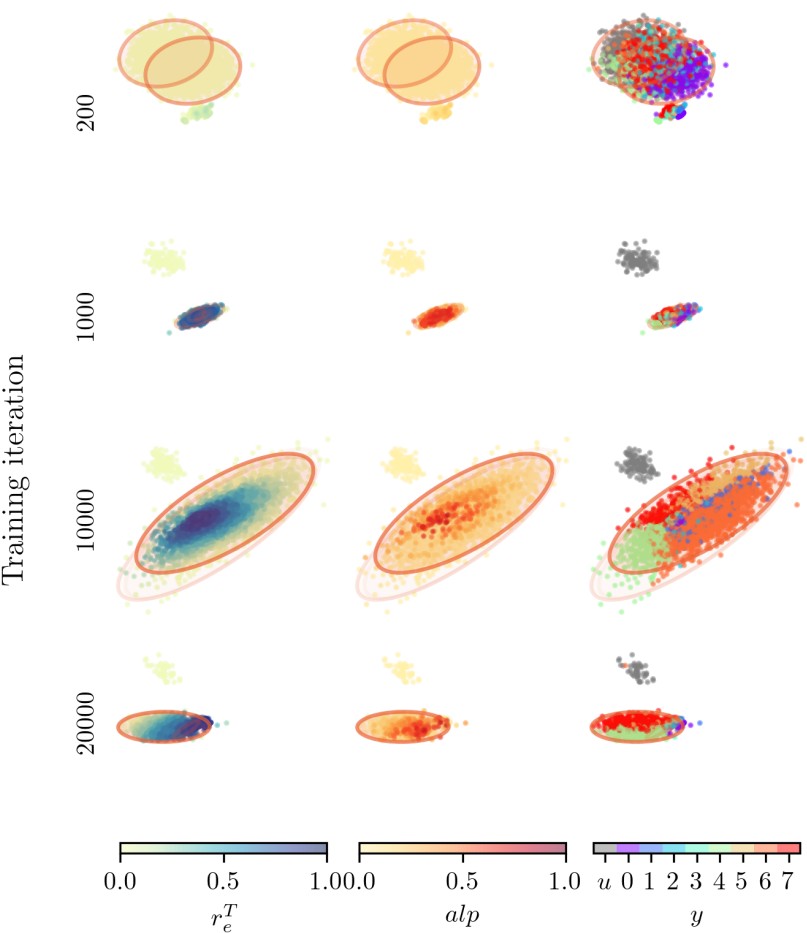

Figure S15: **ALPGMM** sampling strategy migration. Each dot in the visualization corresponds to the latent parameterization of a target motion, with colors representing the policy's tracking performance, the ALP measure, and the predicted motion type. Unlearnable components are colored grey with the label $u$. The ellipsoids represent online GMM clusters used by **ALPGMM**, with the color gradient indicating the mean ALP measure of samples from each cluster.

the latent parameterization space is heavily affected by unlearnable components. In our experiment with $60\%$ unlearnable targets, **ALPGMM** produces the highest tracking performance among all skill samplers by focusing only on regions where the policy excels. The importance of this feature is highlighted in applications that require extracting as many learnable motions as possible from large, high-dimensional target datasets with unknown difficulty distributions.

To further understand the exploration and sampling strategy enabled by **ALPGMM**, we visualize the migration of the sampling region in a representative run with trajectories from an unlearnable motion type in Fig. S15. To this end, we project to 2D plane the latent parameterization samples collected in the skill-performance buffer $\mathcal{B}$, which tracks the running target motions selected by **ALPGMM**, at different stages of training in Fig. S15. For evaluation purposes, an oracle classifier is trained to predict original motion classes $y$ from their latent parameterizations as detailed in Suppl. A.1.2.

Each dot represents the latent parameterization of a target motion and is colored according to the policy's performance $r_e^T$ in tracking this target, the ALP measure $alp$, and the motion type $y$ predicted by the oracle classifier in each column. Specifically, we color the unlearnable components grey with the label $u$ in the motion prediction results. In addition, we plot the online GMM clusters

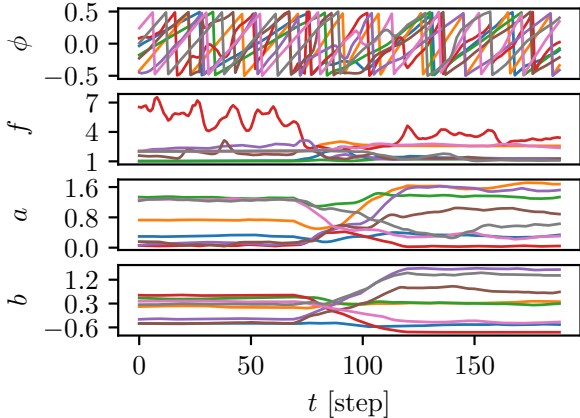

Figure S16: Motion representation of a transition from a periodic motion to another using PAE. The quasi-constant motion parameterization assumption holds only locally.

employed by **ALPGMM** in ellipsoids with the color gradient indicating the mean ALP measure of samples from each cluster.

At the beginning (200 iterations) of learning, motion targets are initialized randomly. The policy is still undertrained at this early stage and thus tracks motions with generally low performance. As the learning proceeds (1000 iterations), the policy gradually improves its capability on the learnable motions indicated by the increased performance and ALP measure on the colored samples. In the meantime, the policy also recognizes the unlearnable region (grey) where it fails to achieve better tracking performance and remains a low learning process. As **ALPGMM** biases its sampling toward Gaussian clusters with high ALP measure, the cluster on the unlearnable motions becomes silent, indicated by an ellipsoid with complete transparency. Further training (10000 iterations) enlarges the coverage of tracking targets indicated by the expanded sampling range centered around the mastered region. A gradient pattern in tracking performance and ALP measure is observed where higher values are achieved at points closer to the confidence region. Finally, more extended training time (20000 iterations) pushes **ALPGMM** to areas more distant from the initial target distribution, motivating the policy to focus on specific motions where the performance may be further improved. This demonstrates the efficacy of **ALPGMM** in navigating the latent parameterization space to focus on learnable regions.

## A.3   LIMITATIONS

### A.3.1   QUASI-CONSTANT MOTION PARAMETERIZATION

FLD provides a solution to representing high-dimensional long-horizon motions in meaningful low dimensions. The training of FLD explicitly enforces latent dynamics that respect spatial-temporal relationships and identify the intrinsic transition patterns in periodic or quasi-periodic motions. However, the propagation of such latent dynamics replies on the quasi-constant motion parameterization assumption (Assump. 1), which holds well on periodic or quasi-periodic motions. When it comes to motions containing less periodicity, we observe time-varying latent parameterization along the trajectory. As depicted in Fig. S16, although the quasi-constant motion parameterization assumption may still hold locally in motions containing aperiodic transitions, the latent dynamics in the transition phase can be challenging to determine. Thus, enforcing a globally constant set of latent parameterization over the whole trajectory is likely to underparameterize the latent embeddings and result in inaccurate reconstruction. If the reconstruction loses too much information with respect to the original motion, additional techniques on period detection and data preprocessing that involve extra human effort have to be applied.

We may also view aperiodic motions as periodic ones with very long periods. This way, the length of the observation window needed to capture the transitions over a whole period is also extended.

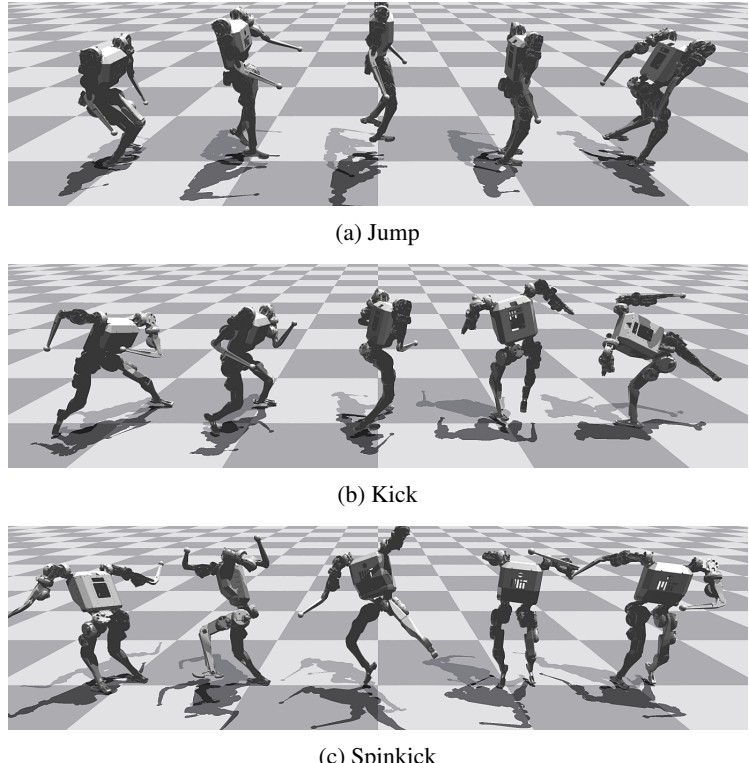

(a) Jump

(b) Kick

(c) Spinkick

Figure S17: Representative motions that require specific reward functions to be learned.

This increases not only the computational complexity but also the reconstruction performance as FLD always performs a global periodic latent approximation of the original motion.

### A.3.2 MOTION-DEPENDENT REWARDS

With an appropriate sampling strategy, FLD is able to continually propose novel tracking targets online that enhance the generality of the learning policy. However, the adaptive curriculum learning methods implemented in this work assume that all such synthesized target motions can be learned with the same set of reward functions. This does not necessarily hold true for large complex datasets where some motions may require specifically designed rewards to be invoked and picked up. Especially for robotic systems, motion execution is strongly constrained by critical physical properties such as non-trivial body part inertia and actuation limits such as motor torques. We exemplify some motions that require different sets of reward functions to be acquired in Fig. S17. Additional efforts in designing target-dependent reward functions are needed to push further the limit of tracking capability of real systems. This may be especially challenging in low-level motor skill development, where such reward functions are strongly correlated with the sensorimotor spaces and physical limits of different embodiments.

### A.3.3 LEARNING-PROGRESS-BASED EXPLORATION

In the adaptive curriculum learning process without privileged access to the offline data, **ALPGMM** searches for motion targets that yield the most significant ALP measure. However, these targets do not necessarily align with the intended reference motions. The policy may end up learning some interpolated motions that do not make sense in terms of motor skills but yield a high ALP measure during training. In these settings, learning-progress-based exploration is allured to regions with the fastest growth in performance, giving up tracking challenging motions where progress requires more extended learning. Occasional random sampling mitigates this problem by reducing exploration's reliance on pure ALP measures. Without access to the offline data, additional careful analysis in locating truly useful motions in the latent parameterization space is needed.

