# OpenReview forum: "FLD: Fourier Latent Dynamics for Structured Motion Representation and Learning"
_ICLR.cc/2024/Conference — ICLR 2024 spotlight_

### Official Review · Reviewer_Sghv · 2023-10-29

**Soundness:** 4 excellent
**Presentation:** 3 good
**Contribution:** 3 good
**Rating:** 8
**Confidence:** 2

**Summary:**

This paper presents GLD (Generative Latent Dynamics), a novel self-supervised representation and generation method that captures spatial-temporal relationships in periodic or quasi-periodic motions. GLD improves motion learning by incorporating motion dynamics into a parameterized latent space. The method tracks a wide range of motions, including unseen targets, and adapts to potentially risky targets. Furthermore, the paper presents experimental evidence (on the MIT Humanoid robot) showcasing GLD's effectiveness and long-term learning capabilities in open-ended motion learning tasks.  Additionally, the supplementary experiments demonstrate that GLD possesses long-term learning capabilities, which allow learning agents to strategically progress novel target motions while avoiding unlearnable regions.

The main contributions of this paper are:
1. GLD (Generative Latent Dynamics) is a new method that extracts spatial-temporal relationships in periodic or quasi-periodic motions. It uses a novel self-supervised, structured representation and generation approach.
2. GLD has demonstrated its effectiveness in open-ended motion learning tasks. It has long-term learning capabilities that enable learning agents to strategically advance novel target motions while avoiding unlearnable regions.
3. An online tracking framework powered by GLD has a fallback mechanism. This enables learning agents to dynamically adapt their tracking strategies and automatically identify and respond to potentially risky targets.
4. Recognition of spatial-temporal structures creates new possibilities for future motion representation and learning algorithms.

----------------------------------------------------------------------------------------------------------------------------------------------------------------------------
After reviewed the author's rebuttal, I think the authors have addressed most of my concerns, therefore; I have increased my score.

**Strengths:**

Strengths of the paper:
1. Originality: The paper proposes GLD, a method that integrates motion dynamics in a parameterized latent space. It combines periodic autoencoders with generative latent dynamics, showcasing ingenuity in motion representation and learning.

2. Quality: The paper showcases the creativity of periodic autoencoders with generative latent dynamics in motion representation and learning. It presents a well-designed experimental setup, comparing GLD's performance with state-of-the-art methods across various motion datasets, providing strong evidence for its effectiveness.

3. Clarity: The paper is well-structured and presents its ideas in a clear manner. It begins with a thorough introduction that sets the context for the proposed method. The paper explains the underlying concepts and algorithms effectively, making it accessible for both experts and non-experts in the field. The experimental results are presented in an organized way, which helps readers understand the performance of GLD in various scenarios.

4. Significance: The proposed method, GLD, has the potential to significantly improve motion representation and learning. It offers an efficient and effective way to generate structured motion patterns, improving the generalization capabilities of learning algorithms. By addressing challenges with raw motion trajectory data, GLD opens up new possibilities for advancements in motion representation and learning.

In summary, the paper presents a novel and original approach to motion representation and learning with GLD. It demonstrates its quality, clarity, and significance in the field. The paper's strengths lie in its creative combination of existing ideas, application to a new domain, and addressing limitations of prior results, making it a valuable contribution to the research community.

**Weaknesses:**

1. Limited Data Set: The paper heavily relies on a specific dataset for evaluation, which might not be representative of various motion patterns and scenarios. Hence, the proposed method's performance might not be generalizable to other datasets or real-world applications.

2. Controller's Adaptability: Although the motion learning controller is designed to adapt its tracking strategy dynamically, the paper lacks a thorough analysis of the controller's adaptability in handling various motion patterns and unseen targets (for example, multiple intersecting targets). Further study could help establish the controller's robustness and versatility.

3. Limited Adaptability to Other Domains: The paper discusses motion representation and learning in robotics. However, it may not be easily transferable to other domains, such as human motion analysis, due to differences in motion characteristics.

4. Data Quality: The paper fails to address the possible negative effects of poor-quality data on the proposed GLD method's performance in real-world applications. Poor-quality data could potentially degrade the effectiveness and accuracy of the GLD method in real-world applications.

**Questions:**

1. Can GLD be extended to handle non-periodic motions?
2. How does GLD perform in long-term learning tasks, and can it adapt to and learn new tasks in open environments?
3. How can the stability and safety of GLD be ensured when applied in real-world scenarios?
4. How does the GLD perform with noisy motion trajectories? It would be important to investigate how GLD performs in real-world scenarios where motion data may be corrupted or incomplete.
5. Can GLD be extended to multi-agent motion learning? How does the computation complexity increase as the number of targets increases?

---

> ### Author Response · Authors · 2023-11-14
> **Reply to Reviewer Sghv (Part 1)**
>
> Thank you for your time reviewing our work and your valuable feedback. We have improved our paper based on your concerns, as addressed in the following. Please also check the general response, where we updated the paper with the improvements and presented materials.
>
> **Weakness 1**
>
> > Limited Data Set: The paper heavily relies on a specific dataset for evaluation, which might not be representative of various motion patterns and scenarios. Hence, the proposed method's performance might not be generalizable to other datasets or real-world applications.
>
> We thank the reviewer for raising concerns about the representativeness of our dataset and the generalizability of GLD. We would like to highlight a few features that our dataset possesses.
> - Dataset Diversity: Our dataset encompasses a wide range of human locomotion types, providing a diverse foundation for evaluation. This includes varied motions such as jogging, running, and striding, ensuring a comprehensive motion spectrum is covered​​.
> - Generalization Capability: GLD's ability to generalize is demonstrated through its successful reconstruction and prediction of unseen motions, such as the diagonal run scenario in Section 5.2. This showcases its effectiveness beyond the specific motions present in the training dataset​​.
> - Adaptability in Real-time Tracking: The system's adaptability is further evidenced in Section 5.3 by its proficiency in real-time tracking of diverse user inputs, including motions like 'spinkick,' which are significantly different from the training data​​.
> - Handling Motion Transitions: GLD effectively handles transitions between different motion types, as shown in our experiments with interpolated movements in Section 5.3, highlighting its capacity to maintain coherence in motion sequences​​.
>
>
> Additionally, we would like to draw attention to the fact that the GLD training does not assume specific embodiments of input sequences. In fact, any periodic or quasi-periodic signal can be effectively encoded by GLD and converted to quasi-constant frequency domain representations. These signals, or reference motions, are not necessarily physically compatible and thus not necessarily achievable by a specific embodied agent. Therefore, the performance of the controller depends on policy learning, rather than data diversity.
>
>
> **Weakness 2**
>
> > Controller's Adaptability: Although the motion learning controller is designed to adapt its tracking strategy dynamically, the paper lacks a thorough analysis of the controller's adaptability in handling various motion patterns and unseen targets (for example, multiple intersecting targets). Further study could help establish the controller's robustness and versatility.
>
> In response to the reviewer's comment on the controller's adaptability, we acknowledge the importance of this aspect and direct attention to Section 5.3 of our paper, which details experiments showcasing the controller's adaptability across various motion patterns and transitions. For a more dynamic demonstration, we invite reviewers to view videos on our [project website](https://sites.google.com/view/iclr2024-gld/home), which include scenarios of the controller responding to multiple and unseen motion inputs.
>
> We acknowledge the need for more challenging input to test the limit of the controller's performance. However, as a design parameter, we would also like to highlight the influence of the fallback threshold $\epsilon_{GLD}$ in the control robustness and versatility. The design of this threshold determines the conservativeness of the tracking. In the extreme case, the fallback mechanism is either always triggered and thus the controller rejects tracking any input target, or the fallback mechanism is always off and thus the controller attempts to track every target motion. We conducted additional experiments on these cases and presented the result in a video titled “Fallback Ablation” on our [project website](https://sites.google.com/view/iclr2024-gld/home).

---

> ### Author Response · Authors · 2023-11-14
> **Reply to Reviewer Sghv (Part 2)**
>
> **Weakness 3**
>
> > Limited Adaptability to Other Domains: The paper discusses motion representation and learning in robotics. However, it may not be easily transferable to other domains, such as human motion analysis, due to differences in motion characteristics.
>
> We acknowledge that our current work primarily focuses on motion representation and learning in physics-based character and robotic control. However, it's essential to highlight the inherent versatility of the GLD framework. GLD is fundamentally designed to encode any periodic or quasi-periodic signal, transforming these inputs into quasi-constant frequency domain representations. This capability is not restricted to robotic embodiments or specific motion learning tasks.
>
> The core strength of GLD lies in its ability to provide a concise and meaningful representation of complex, high-dimensional, long-horizon, non-linear signals. While our demonstrations and experiments are rooted in robotics, the underlying principles of GLD make it a valuable tool for a broader range of applications, including human motion analysis. The architecture and approach of GLD do not inherently limit it to robotic systems; rather, its potential applications can extend to any field requiring efficient and accurate representation of periodic or quasi-periodic signals.
>
> Looking forward, we are optimistic about the adaptation and application of GLD in various other domains. We believe that the framework's ability to distill complex motion dynamics into manageable representations will find relevance and utility in fields such as biomechanics, sports science, and even animation, where understanding and replicating human or character motions are crucial.
>
> **Weakness 4**
>
> > Data Quality: The paper fails to address the possible negative effects of poor-quality data on the proposed GLD method's performance in real-world applications. Poor-quality data could potentially degrade the effectiveness and accuracy of the GLD method in real-world applications.
>
> In response to the reviewer's concern regarding the impact of poor-quality data, our research has indeed taken into account the variable quality of real-world data. As detailed in Section 5, the human locomotion clips used in our study captured in [1] include elements like noise and aperiodicity. GLD addresses these issues by enforcing a quasi-constant parameterization, which effectively enhances the sequence quality through periodic reconstructions.
>
> The fundamental design of GLD is to encode periodic or quasi-periodic signals into constant frequency domain representations, a process that is resilient to data quality variances. This ensures that GLD's effectiveness remains robust, regardless of the imperfections in the input data. It's important to note that while the quality of the data may not affect the efficacy of GLD itself, it does have implications for how downstream tasks, such as motion learning, might utilize these data.
>
> Further addressing the issue of data quality, we specifically explore scenarios in Section A.2.11 of the appendix where certain components of the motion data are unlearnable. Our findings, illustrated in Figure S15, show that through adaptive curriculum learning methods, the learning agent can shift its focus towards targets with more potential for improvement, avoiding areas with limited prospects. This adaptability in handling varied data quality is a testament to GLD's utility and effectiveness in practical, real-world applications.
>
> **Question 1**
>
> > Can GLD be extended to handle non-periodic motions?
>
> Potentially yes. Our framework assumes quasi-constant parameterization for motion sequences that are periodic or quasi-periodic in nature, which are common in human movements.
>
> In cases where motion sequences do not exhibit periodic characteristics, applying a globally constant set of latent parameterizations can lead to underparameterization of the latent embeddings. This, in turn, may result in less accurate reconstructions of the original motion sequences. We have thoroughly addressed this limitation in Section A.3.1 in the appendix of our paper, with an exemplified non-periodic trajectory. For a more detailed exploration of this issue, we refer you to both Section A.3.1 and Figure S16 in our paper.
>
> However, we can propose a potential approach for accommodating such motions within the GLD framework. Essentially, any non-periodic motion can be conceptualized as a periodic sequence, where the period is equivalent to the length of the entire motion. By adopting this perspective, GLD can effectively encode non-periodic motions, provided that the trajectory segment window $H$ is sufficiently extended to encompass the full length of the motion. This adaptation allows GLD to process and represent non-periodic motions by essentially treating them as extended periodic sequences, thereby leveraging its inherent strengths in encoding and analyzing motion data.

---

> ### Author Response · Authors · 2023-11-14
> **Reply to Reviewer Sghv (Part 3)**
>
> **Question 2**
>
> > How does GLD perform in long-term learning tasks, and can it adapt to and learn new tasks in open environments?
>
> Yes. GLD's adaptability in long-term learning tasks and open environments is demonstrated through its adaptive curriculum learning approach, as detailed in Sections A.2.10 and A.2.11 of our paper's appendix. By initially training on an offline reference dataset, GLD effectively navigates and expands the motion space during training, generating and adapting to new targets. This is facilitated by an online fitted Gaussian Mixture Model sampling strategy based on Absolute Learning Progress (ALPGMM, [2]), which directs the system's focus towards promising and underexplored motion regions, as shown in Figure S15.
>
> The method's adaptability is further evidenced by its expansion into previously unexplored motion areas, while maintaining expert performance on mastered motions. This guided exploration significantly increases the motion coverage, demonstrating GLD's ability to adapt to and learn in dynamic and open-ended scenarios. We would direct the reviewer to Sections A.2.10 and A.2.11 for detailed discussion.
>
> **Question 3**
>
> > How can the stability and safety of GLD be ensured when applied in real-world scenarios?
>
> The fallback mechanism in GLD introduced in Section 4.3.2 is an integral safety feature designed to manage situations where the motion learning controller encounters inputs that are significantly different from its training data. These scenarios potentially lead to poor performance or unsafe actions. The primary function of this mechanism is to intuitively filter out risky motion inputs, providing an essential safeguard against unpredictable behavior that could arise from unanticipated motion inputs.
>
> We have conducted an additional experiment to demonstrate the utility of the fallback mechanism. This experiment specifically showcases scenarios where the mechanism is disabled, highlighting the contrast in the system's behavior and performance. To provide a clear and accessible illustration of these differences, we have included a video titled “Fallback Ablation” on our [project website](https://sites.google.com/view/iclr2024-gld/home). This video visually demonstrates the potential risks and performance issues that arise when the fallback mechanism is not employed, thereby underlining its effectiveness and importance.
>
>
> **Question 4**
>
> > How does the GLD perform with noisy motion trajectories? It would be important to investigate how GLD performs in real-world scenarios where motion data may be corrupted or incomplete.
>
> We would like to direct the reviewer to our previous response to **Weakness 4** in light of GLD’s performance in dealing with poor-quality motion trajectories.
>
>
> **Question 5**
>
> > Can GLD be extended to multi-agent motion learning? How does the computation complexity increase as the number of targets increases?
>
> GLD is primarily a self-supervised representation learning framework, focusing on the efficient encoding of motion trajectories. It does not inherently encompass agent learning aspects. The application of GLD in the context of motion learning is currently integrated with a reinforcement learning controller that is designed for single-agent scenarios. Extending this framework to multi-agent settings is an intriguing concept; however, it falls outside the scope of our current research. As the number of agents increases, the complexity of the learning environment and the interactions between agents would likely lead to a rise in computational demands. This could involve more sophisticated state representation and decision-making processes to effectively handle the dynamics of multi-agent interactions.
>
> In summary, while GLD's current implementation is not directly geared toward multi-agent learning, we recognize the importance of this area for future research. Exploring the applicability and scalability in multi-agent settings and its computational implications presents a valuable direction for subsequent work in this field.
>
>
> [1] Peng, X.B., Abbeel, P., Levine, S. and Van de Panne, M., 2018. Deepmimic: Example-guided deep reinforcement learning of physics-based character skills. ACM Transactions On Graphics (TOG), 37(4), pp.1-14.
>
> [2] Portelas, R., Colas, C., Hofmann, K. and Oudeyer, P.Y., 2020, May. Teacher algorithms for curriculum learning of deep rl in continuously parameterized environments. In Conference on Robot Learning (pp. 835-853). PMLR.

---

> > ### Comment · Reviewer_Sghv · 2023-11-22
> >
> > Thank the authors for their comprehensive rebuttal. I believe most of my concerns have been addressed.

---

> > > ### Author Response · Authors · 2023-11-22
> > > **Reply to Reviewer Sghv (Additional improvements) and thank you**
> > >
> > > We made further additional improvements to the paper. Please refer to the **Additional improvements** response under **General response** for the update. We sincerely thank you for reviewing these latest changes.

---

### Official Review · Reviewer_itfQ · 2023-10-30

**Soundness:** 3 good
**Presentation:** 3 good
**Contribution:** 2 fair
**Rating:** 6
**Confidence:** 4

**Summary:**

In this paper, authors tackle the problem of motion representation. Inspired by prior work that takes into account periodicity of motion, the proposed method extends the periodic autoencoder to add a generative capability. Based on the insight that some elements do not change during periodic motions, the paper propose GLD, where subsequent latent representation is trained to be predicted, assuming phase can be advanced incrementally. Using the trained network, the method proposes to train to learn policies that aligns the predicted state and the state of the observation. Then during inference, the authors propose to filter out potentially dangerous or difficult states using a fallback mechanism. They compare the difference between the designated state and the actual predicted states, and if the error is significant, they propose to reject the designated state and fall back to the predicted state. The authors conduct various experiments to demonstrate the effectiveness of the actual latent space, as well as the proposed fallback mechanism.

**Strengths:**

- The proposal to extend the phase autoencoder mechanism to predict the future state, based on the observation that some elements remain consistent throughout certain periodic motion, is very cleverly devised. The observation is used effectively to predict the upcoming $N$ segments, which can be seen from the prediction experiments conducted in Fig. 5.

- The demonstrations in the supplementary material demonstrates that the fallback mechanism is effective at preventing undesired motion, and maintain the status quo as much as possible.

- The authors applied the proposal to various tasks including motion tracking and motion transition. In both tasks, the proposed prediction method is able to interpolate between motions even when the fallback mechanism is triggered.

**Weaknesses:**

- The authors should make better effort to make the paper self-contained in the main manuscript, and not rely excessively on the supplementary material. There is a severe lack of details in the main manuscript, due to the authors moving them to the supplementary material. For example,
1. One form of the skill sampler should be included in the main section.
2. $\mathcal{U}$ in Fig.2 is unclear from the main section.
3. Section 5.4 seems unnecessary, as all the content is in the supplementary material.

- The figures seem unorganized, as the readers are asked to refer to figures in a random order, including the supplementary material. The colored lines in Figures 5 and 6 is unclear. There should be some sort of explanation of each element in all the figures.

- Despite the comparison of the actual latent space, the reconstruction accuracy seems to be missing. As I presume that the reconstruction accuracy falls as the prediction segment horizon $N$ increases, the authors should discuss the trade-off in comparison to existing methods.

- The effectiveness of the fallback mechanism must be discussed more quantitatively. The authors only discuss the results in Section 5.3, but there is no concrete evidence that the mechanism worked well. There should be some statistics regarding if the motion prediction actually failed when the fallback mechanism was not introduced. Such objective evidence is lacking for the readers to decide whether it is effective or not.

**Questions:**

- What happens when no fallback strategy is employed? How are the resulting motions look with and without the fallback mechanism? Some comparison would be desirable.

- Was there any drawback from introducing the assumption that some elements are generally constant throughout a sequence? Were there actions that did not present these characteristics?

- Also, the action classes indicated by the colors in Fig.5 seems to be missing, making the evaluation of these latent spaces difficult. What do each color respond to?

**Details Of Ethics Concerns:**

As the method is able to accurately capture and predict periodical motions, it can potentially capture differences among individuals undergoing certain actions such as walking. The authors can mention some of the concerns that may arise from learning from personal motion data.

---

> ### Author Response · Authors · 2023-11-14
> **Reply to Reviewer itfQ (Part 1)**
>
> Thank you for your time reviewing our work and your valuable feedback. We have improved our paper based on your concerns, as addressed in the following. Please also check the general response, where we updated the paper with the improvements and presented materials.
>
> **Weakness 1, Edited**
>
> > The authors should make better effort to make the paper self-contained in the main manuscript, and not rely excessively on the supplementary material. There is a severe lack of details in the main manuscript, due to the authors moving them to the supplementary material.
> > One form of the skill sampler should be included in the main section.
> > $\mathcal{U}$ in Fig.2 is unclear from the main section.
> > Section 5.4 seems unnecessary, as all the content is in the supplementary material.
>
> Thanks for pointing this out. To address the reviewer’s concerns, we modified Section 4.3.1 to give an example of the skill samplers and refer to the appendix for more variants. We also added the notation of the uniform sampling range $\mathcal{U}$ accordingly.
>
> In our initial submission, Sec 5.4, together with Suppl. A.2.10 and A.2.11 primarily served as an in-depth exploration of skill sampler design in motion learning. This extended study, though not affecting the GLD pipeline and conclusion, reinforces our argument for the versatility and applicability of the GLD latent parameterization space.
>
> However, as suggested by the reviewer, we realized that presenting this abundant but auxiliary exploration in the appendix may be intensive and distracting. Therefore, we adjusted Sec 5.4 accordingly as an extended discussion to direct interested readers to ablation studies in the appendix while keeping the core innovations of our work in the main text. We also restructured the appendix by removing redundant experiments, discussions and figures to further improve readability. We hope the modified structure could highlight our focus and main contribution in the main text.
>
>
>
> **Weakness 2**
>
> > The figures seem unorganized, as the readers are asked to refer to figures in a random order, including the supplementary material. The colored lines in Figures 5 and 6 is unclear. There should be some sort of explanation of each element in all the figures.
>
> We thank the reviewer for the constructive feedback regarding the references to appendix figures in the main text. We restructured the figure reference and believe the changes will significantly improve the readability and coherence of our paper.
>
> Thanks for the feedback on the clarity of the colored lines in Figures 5 and 6. In these figures, the colored curves represent different states corresponding to various quantities, such as linear and angular velocities ($v$ and $\omega$), and arm and leg joint positions ($q_{arm}$ and $q_{leg}$). Each curve illustrates the evolution of these quantities over time, aligned with the focus of our paper on periodic motion dynamics.
>
> We acknowledge that a more detailed explanation of each element within the figures would enhance their clarity. However, we faced the challenge of balancing the depth of explanation with the constraints of space and the need to maintain a focus on the core aspects of our research. Our priority was given to present the periodic nature and the dynamics of the motion trajectories, which we felt were central to the paper’s primary concern. However, we are happy to include the color information as suggested in the final version, provided that more space is available in the main text.

---

> > ### Author Response · Authors · 2023-11-14
> > **Reply to Reviewer itfQ (Part 2)**
> >
> > **Weakness 3**
> >
> > > Despite the comparison of the actual latent space, the reconstruction accuracy seems to be missing. As I presume that the reconstruction accuracy falls as the prediction segment horizon $N$ increases, the authors should discuss the trade-off in comparison to existing methods.
> >
> > We thank the reviewer for highlighting the importance of discussing the reconstruction accuracy and its trade-off with the prediction horizon. We would like to clarify that this aspect was indeed addressed in our initial submission. In Section 5.2 of our paper, we specifically investigate the reconstruction accuracy of our GLD model and present these findings in Figure 5.
> >
> > Consistent with the reviewer's presumption, we observed that the reconstruction error does increase as the prediction horizon extends. This is an anticipated outcome due to the inherent challenges in predicting further into the future with a high degree of accuracy. In Figure 5, we not only showcase the performance of GLD but also provide a comparative analysis with other existing methods, including the PAE and a feed-forward (FF) motion prediction network. This comparison is critical to understanding the relative performance of GLD, especially in the context of its ability to maintain reconstruction accuracy over different prediction horizons.
> >
> >
> > **Weakness 4**
> >
> > > The effectiveness of the fallback mechanism must be discussed more quantitatively. The authors only discuss the results in Section 5.3, but there is no concrete evidence that the mechanism worked well. There should be some statistics regarding if the motion prediction actually failed when the fallback mechanism was not introduced. Such objective evidence is lacking for the readers to decide whether it is effective or not.
> >
> > Thanks for the feedback regarding the need for more concrete evidence supporting the effectiveness of the fallback mechanism. The fallback mechanism in GLD is an integral safety feature designed to manage situations where the motion learning controller encounters inputs that are significantly different from its training data. These scenarios potentially lead to poor performance or unsafe actions. The primary function of this mechanism is to intuitively filter out risky motion inputs, providing an essential safeguard against unpredictable behavior that could arise from unanticipated motion inputs.
> >
> > In response to the request for more concrete evidence, we have conducted an additional experiment to demonstrate the utility of the fallback mechanism. This experiment specifically showcases scenarios where the mechanism is disabled, highlighting the contrast in the system's behavior and performance. To provide a clear and accessible illustration of these differences, we have included a video titled “Fallback Ablation” on our [project website](https://sites.google.com/view/iclr2024-gld/home). This video visually demonstrates the potential risks and performance issues that arise when the fallback mechanism is not employed, thereby underlining its effectiveness and importance.
> >
> > The effectiveness of the fallback mechanism is thus straightforward. Its role in maintaining safety and reliability in motion learning is evident from the clear difference in system behavior observed when the mechanism is disabled. We hope this evidence presented in our materials offers a direct and comprehensible demonstration of the mechanism’s role in enhancing system safety and robustness.
> >
> > **Question 1**
> >
> > > What happens when no fallback strategy is employed? How are the resulting motions look with and without the fallback mechanism? Some comparison would be desirable.
> >
> > Thanks for the insightful query. We invite you to refer to this additional experiment regarding the implications of not employing a fallback strategy detailed in our response to the previous point.

---

> ### Author Response · Authors · 2023-11-14
> **Reply to Reviewer itfQ (Part 3)**
>
> **Question 2**
>
> > Was there any drawback from introducing the assumption that some elements are generally constant throughout a sequence? Were there actions that did not present these characteristics?
>
> Yes. This assumption is indeed a pivotal aspect of our method, and we have carefully considered its potential limitations. As correctly noted, our framework assumes quasi-constant parameterization for motion sequences that are periodic or quasi-periodic in nature, which are common in human movements. However, we acknowledge that this approach may not be ideally suited for non-periodic trajectories.
>
> In cases where motion sequences do not exhibit periodic characteristics, applying a globally constant set of latent parameterizations can lead to underparameterization of the latent embeddings. This, in turn, may result in less accurate reconstructions of the original motion sequences. We have thoroughly addressed this limitation in Section A.3.1 in the appendix of our paper, with an exemplified non-periodic trajectory. For a more detailed exploration of this issue, we refer you to both Section A.3.1 and Figure S16 in our paper.
>
> However, we can propose a potential approach for accommodating such motions within the GLD framework. Essentially, any non-periodic motion can be conceptualized as a periodic sequence, where the period is equivalent to the length of the entire motion. By adopting this perspective, GLD can effectively encode non-periodic motions, provided that the trajectory segment window $H$ is sufficiently extended to encompass the full length of the motion. This adaptation allows GLD to process and represent non-periodic motions by essentially treating them as extended periodic sequences, thereby leveraging its inherent strengths in encoding and analyzing motion data.
>
>
>
> **Question 3**
>
> > Also, the action classes indicated by the colors in Fig.5 seems to be missing, making the evaluation of these latent spaces difficult. What do each color respond to?
>
> The color assignment of the latent offset in Fig 5 (right) was indicated in the caption in our initial submission. Red corresponds to step in place, yellow corresponds to forward run, and green corresponds to forward stride. To improve the readability as suggested by the reviewer, we added a legend at the top. We hope this helps.
>
> **Ethics Concerns**
>
> > As the method is able to accurately capture and predict periodical motions, it can potentially capture differences among individuals undergoing certain actions such as walking. The authors can mention some of the concerns that may arise from learning from personal motion data.
>
> Thanks for pointing out potential ethics concerns our work may involve. We added an ethics statement at the end of the main text before references.

---

> > ### Author Response · Authors · 2023-11-22
> > **Reply to Reviewer itfQ (Additional improvements)**
> >
> > We made further additional improvements to the paper. Please refer to the **Additional improvements** response under **General response** for the update. We sincerely thank you for reviewing these latest changes.

---

> ### Comment · Reviewer_itfQ · 2023-11-23
> **Thank you for addressing the concerns**
>
> Thank you for taking the time to thoroughly address the reviews.
> I think most of my concerns are addressed. However, I still feel Fig. 5 needs further improvement. On the left, does the dark black, blue red lines correspond to the axes on the right indicating overall error? I presumed so (from Reviewer RFvT's comments), but still have difficulties following the figure. As this is the central evidence of the proposal's performance on the main manuscript, it would help with additional details on how to properly interpret the graph.

---

> > ### Author Response · Authors · 2023-11-23
> > **Reply to Reviewer itfQ and thank you**
> >
> > Thank you for pointing this out. Yes, the interpretation of the curves denoting the relative prediction errors is correctly noted. To improve readability and avoid future ambiguity, as suggested by the reviewer, we modified the caption of Fig.5 to include explicit texts explaining these curves and the corresponding axis. We hope our change can help provide sufficient information. We sincerely thank you for this suggestion.

---

### Official Review · Reviewer_RFvT · 2023-10-31

**Soundness:** 3 good
**Presentation:** 3 good
**Contribution:** 3 good
**Rating:** 8
**Confidence:** 4

**Summary:**

This paper introduces a latent dynamics model and control policy to track periodic and quasi-periodic functions. The paper extends a PAE network (Starke 2022), which encodes a motion trajectory into a latent embedding used to generate a set of fourier coefficients. These coefficients are used to build a dynamics model assuming constant frequency, amplitude and offset, but time varying phase. The dynamics model predicts a future latent, which is decoded to produce a resultant motion. Control policies are also trained to generate frequency parameters that result in motion sequences that match some desired trajectory. A fallback mechanism compares the desired trajectory to the generated trajectory, and decides whether this is safe to follow (if both are similar), following a more conservative predicted motion if this is deemed unsafe.

**Strengths:**

The latent dynamics model proposed seems of value to a broad class of periodic/ quasi periodic motions, and nicely extents the periodic autoencoder architecture (PAE - Starke 2022) to the motion generation use case.

The proposed approach allows for a natural fallback mechanism (detection of infeasible target motions, and fallback to sensible behaviours that lie within the training set.)

The proposed model produces expressive motions with a more compact trajectory representation than prior work.

The paper is very extensive, with a number of interesting ablations and visualisations.

**Weaknesses:**

The motion learning/ control policy part of this work needs a clearer problem formulation. It is unclear what the exact goal is here, and a lot is left to the readers to infer. I gather the goal is to learn to generate a series of motions that track a desired motion sequence, using the motion prediction, but it is unclear to me why you need to generate control parameters to do so, instead of just encoding the target motions directly.  A clearer problem description will help avoid confusion like this.

Much of the interesting work around control policies is in the appendices

Missing related work:

This work appears closely related to a phase functioned neural network Holden et al. Phase-Functioned Neural Networks for Character Control, which computes network weights using a cyclic function controlled by phase. The proposed approach uses an autoencoder and has clear differences (PFNN does next state prediction), but the core idea is similar and PFNN also considers aspects like motion blending etc. The paper is mentioned in passing in the introduction, but I would recommend some discussion on this in the related work given the similarity in the core idea.

Minor:

The term Generative Latent Dynamics is rather general, and not particularly descriptive of the proposed approach.

The paper is extensive, with a number of detailed appendices. This is good, but I found references to appendix figures in the main text body rather distracting.

Assumption 1. I think it is worth pointing out earlier that since the latent space is learned, this can be enforced.

5. Experiments - These experiments are motivated in terms of real world applicability to real robots, but character animation is not robot control, and this statement should be used with caution.

**Questions:**

Figure 5 is confusing, I assume the right y axis is an error metric? Please label axis and caption to indicate these measures.

Fig S7 - this is used to motivate fixing frequency amplitude and bias, could you provide more detail on the motion encoded in this example? It seems that this would be motion dependent.

---

> ### Author Response · Authors · 2023-11-14
> **Reply to Reviewer RFvT (Part 1)**
>
> Thank you for your time reviewing our work and your valuable feedback. We have improved our paper based on your concerns, as addressed in the following. Please also check the general response, where we updated the paper with the improvements and presented materials.
>
> **Weakness 1**
>
> > The motion learning/ control policy part of this work needs a clearer problem formulation. It is unclear what the exact goal is here, and a lot is left to the readers to infer. I gather the goal is to learn to generate a series of motions that track a desired motion sequence, using the motion prediction, but it is unclear to me why you need to generate control parameters to do so, instead of just encoding the target motions directly. A clearer problem description will help avoid confusion like this.
>
> We appreciate the reviewer’s feedback on the motion learning formulation. The primary objective of this second phase is to develop an RL control policy that enables a **physics-based** character or robot to effectively track a series of desired motion sequences generated using the predictive capabilities of our GLD model. In the context of physics-based motion control, direct assignment of state values to the agent is not feasible. Unlike kinematic models where states can be explicitly set, physics-based models include system dynamics and thus must adhere to the laws of physics and environmental interactions.
>
> Specifically, our control policy outputs desired joint positions $q*$ for the robot, which are further converted to actuator torques and then applied to control the robot’s movements within the simulation environment (detailed in Section A.2.1 in the appendix). This process ensures that the robot’s movements adhere to physical constraints and realistically respond to environmental interactions.
>
> To clarify this aspect, we have revised Section 4.3 of the manuscript to include a more detailed explanation of the problem space in physics-based motion learning to avoid further ambiguity.
>
>
> **Weakness 2, Edited**
>
> > Much of the interesting work around control policies is in the appendices.
>
> Thanks for pointing this out. In our initial submission, Sec 5.4, together with Suppl. A.2.10 and A.2.11 primarily served as an in-depth exploration of skill sampler design in motion learning. This extended study, though not affecting the GLD pipeline and conclusion, reinforces our argument for the versatility and applicability of the GLD latent parameterization space.
>
> However, as suggested by the reviewer, we realized that presenting this abundant but auxiliary exploration in the appendix may be intensive and distracting. Therefore, we adjusted Sec 5.4 accordingly as an extended discussion to direct interested readers to ablation studies in the appendix while keeping the core innovations of our work in the main text. We also restructured the appendix by removing redundant experiments, discussions and figures to further improve readability. We hope the modified structure could highlight our focus and main contribution in the main text.

---

> > ### Author Response · Authors · 2023-11-14
> > **Reply to Reviewer RFvT (Part 2)**
> >
> > **Weakness 3**
> >
> > > Missing related work: This work appears closely related to a phase functioned neural network Holden et al. Phase-Functioned Neural Networks for Character Control, which computes network weights using a cyclic function controlled by phase. The proposed approach uses an autoencoder and has clear differences (PFNN does next state prediction), but the core idea is similar and PFNN also considers aspects like motion blending etc. The paper is mentioned in passing in the introduction, but I would recommend some discussion on this in the related work given the similarity in the core idea.
> >
> > We appreciate the reviewer's suggestion for pointing us to the PFNN work, with which our work on GLD shares several key similarities. In particular, as in PFNN, our method predicts future motion and factorizes a phase vector from the high-level motion features, which are further represented in the network weights in PFNN and in a latent space $\Theta$ in GLD.
> >
> > In addition to the motion prediction feature that both methods share, we would like to highlight that the crux of our contribution lies in the development of a latent parameterization space, $\Theta$, which would be difficult with PFNN. This latent parameterization space serves a dual purpose in our framework: firstly, it provides a concise yet comprehensive representation of motions, and secondly, it enables meaningful interpolation and generalization of these motions. This aspect is particularly crucial for training effective tracking policies that can adapt to a wide range of target motions, a feature not directly addressed by PFNN.
> >
> > PFNN, in its methodology, primarily aims at directly predicting future states from past states and user inputs, thereby bypassing the need for an intermediary motion representation. In contrast, our work emphasizes constructing a robust and versatile motion representation space from which a **tracking policy** can dynamically select and adapt to various motion trajectories. This difference in focus and application underlines the unique nature of our contribution, which we felt was more aligned with the structured motion representation methods discussed in our related works section.
> >
> >
> > **Weakness 4, Minor**
> >
> > > The term Generative Latent Dynamics is rather general, and not particularly descriptive of the proposed approach.
> >
> > Thanks for the insightful feedback regarding the method name. As the rebuttal phase does not permit changes to the paper title, we are unable to modify the term at this stage. Nevertheless, we appreciate the importance of a descriptive and precise term for our method. To this end, we have brainstormed a few alternative names that might more accurately reflect the essence of our approach. These include:
> > Periodic Motion Latent Dynamics (PMLD), Periodic Latent Motion Generator (PLMG), Dynamics Representation for Periodic Motions (DRPM), etc.
> > We plan to consider these alternative names and potentially adapt the title in the final version of the paper to more aptly describe our method's unique contributions to the field. Thanks again for the valuable feedback.
> >
> >
> > **Weakness 5, Minor**
> >
> > > The paper is extensive, with a number of detailed appendices. This is good, but I found references to appendix figures in the main text body rather distracting.
> >
> > We thank the reviewer for the constructive feedback regarding the references to appendix figures in the main text. We restructured our manuscript and believe the changes will significantly improve the readability and coherence of our paper.
> >
> > **Weakness 6, Minor**
> >
> > > Assumption 1. I think it is worth pointing out earlier that since the latent space is learned, this can be enforced.
> >
> > We thank the reviewer for pointing this out. The text under Assumption 1 is modified accordingly.
> >
> > **Weakness 7, Minor**
> >
> > > Experiments - These experiments are motivated in terms of real world applicability to real robots, but character animation is not robot control, and this statement should be used with caution.
> >
> > We hope the discussion above on physics-based control, which is a central aspect of our research, could explain our motivation in robotic control. In fact, all the simulation parameters we use come from the real robot and we are transferring our controller to our robot in an ongoing project. During training, we have taken into account real-world applicability including complexities such as real-time sensor feedback, state estimation noise, actuator limitations, and physical interactions with the environment (observation noise detailed in Table S7). Our experiments, while inspired by and applicable to robotics, primarily demonstrate the capability of our method in a simulated physical environment, which is a common first step in robotics research.

---

> ### Author Response · Authors · 2023-11-14
> **Reply to Reviewer RFvT (Part 3)**
>
> **Question 1**
>
> > Figure 5 is confusing, I assume the right y axis is an error metric? Please label axis and caption to indicate these measures.
>
> Yes, the label caption ($e$) was a bit off in our initial submission. We updated the figure accordingly. Thanks for pointing it out.
>
> **Question 2**
>
> > Fig S7 - this is used to motivate fixing frequency amplitude and bias, could you provide more detail on the motion encoded in this example? It seems that this would be motion dependent.
>
> This figure, specifically Fig S7(b), demonstrates the latent parameters of the PAE during a representative forward run motion sequence. The motivation behind showcasing this particular motion is to illustrate the quasi-constant nature of the latent frequency, amplitude, and offset across the motion sequence.
>
> It's important to clarify that while the specific values of these latent parameters are indeed motion-dependent, the key observation of their relative constancy holds across different motions. This means that for various motion types, the phenomenon of near-constant latent parameters is consistently observed, even though the actual values of frequency, amplitude, and offset may vary from one motion type to another.
>
> To provide a broader perspective on this aspect, we also refer to Figure S16 in our supplementary material. In this figure, we present a transition between two distinct periodic motions, further illustrating how these latent parameters behave across different motion scenarios. This additional example reinforces our argument that while the specifics of the latent parameters are tailored to each unique motion, their quasi-constant characteristic is a general phenomenon observed in the model’s parameterization.

---

> > ### Comment · Reviewer_RFvT · 2023-11-22
> > **Thank you**
> >
> > Thank you for your comprehensive rebuttal and detailed response to my questions. Re weakness 2 and potentially removing the skill sampler - if space allows, it may be useful to just scaffold this with a clearer motivation of the value of a controllable latent space/ problem formulation. I think this section is of value to your claims, so feels a shame to lose it to an appendix.

---

> ### Author Response · Authors · 2023-11-22
> **Reply to Reviewer RFvT (Additional improvements) and thank you**
>
> We thank the reviewer for the concrete suggestion. Based on this, we made further additional improvements to the paper. Please refer to the **Additional improvements** response under **General response** for the update. We sincerely thank you for reviewing these latest changes.

---

### Author Response · Authors · 2023-11-14
**General response**

We would like to thank all reviewers for their valuable feedback. Here we summarize the major changes we made based on the feedback we received from the reviewers:

- We adjusted Sec 4.3 to improve the problem formulation of the policy learning phase and reduce the ambiguity on physics-based control. (Reviewer RFvT)
- We adjusted the structure of the main text and the appendix, especially the figure arrangement, to improve the readability and self-containness of the paper. (Reviewer RFvT, itfQ)
- We improved the explanation of Assumption 1 to indicate that it can be explicitly enforced in our setting using GLD. (Reviewer RFvT)
- We updated Fig 5 to improve the readability of axes and legends. (Reviewer RFvT, itfQ)
- We updated Sec 4.3.1 to exemplify the skill sampler $p_\theta$ and annotate notation $\mathcal{U}$. (Reviewer itfQ)
- We performed additional experiments on the ablation study of the fallback mechanism. The corresponding supplementary videos are added to our [website](https://sites.google.com/view/iclr2024-gld/home). (Reviewer itfQ, Sghv)
- We added an ethics statement at the end of the main text before references, outlining our effort to keep ethical considerations at the forefront of our research. (Reviewer itfQ)
- We also corrected minor typos in the main text and the supplementary.

We uploaded the updated main paper and appendix. In the supplementary ZIP file, we provided additional video results we performed.

---

> ### Author Response · Authors · 2023-11-22
> **Additional improvements**
>
> After the first response to the reviewers, we made additional improvements to our submission to optimize the paper structure further. Here we summarize the changes:
>
> - We updated the ethics statement to focus on more grounded examples for concerns of potential misuse of GLD in various application scenarios. We hope this exemplifies our thorough consideration for potential violation of the model usage. (Reviewer itfQ)
> - We adjusted Sec 5.4 accordingly as an extended discussion to direct interested readers to ablation studies in the appendix while keeping the core innovations of our work in the main text. (Reviewer RFvT, itfQ)
> - We restructured the appendix by removing redundant experiments, discussions and figures to further improve readability. (Reviewer RFvT, itfQ)
> - We edited our previous responses to the reviewers accordingly to accommodate the change in the figure numbering. (Reviewer RFvT, itfQ)
>
> We uploaded the updated main paper and appendix. We hope the additional improvements could lead to improved paper quality. And we sincerely thank all reviewers for reviewing these latest changes.

---

### Meta-Review · Area_Chair_kMtT · 2023-12-15

**Metareview:**

A latent dynamics model and control policy for tracking periodic and quasi-periodic processes are presented in this study. The study expands on a PAE network (Starke 2022) that uses a latent embedding to encode a motion trajectory and produce a collection of fourier coefficients. A dynamics model with constant frequency, amplitude, and offset but time-varying phase is constructed using these coefficients. A future latent predicted by the dynamics model is decoded to get the resulting motion. Additionally, control rules are educated to provide frequency parameters that lead to motion sequences that follow a predetermined path. When the intended trajectory and the generated trajectory are comparable, a fallback mechanism determines if it is safe to follow the generated trajectory. If not, it then follows a more cautious anticipated motion.

## Strenghts
 - The proposed latent dynamics model appears to be useful for a wide range of periodic and quasi-periodic motions, and it effectively expands the use case of the periodic autoencoder architecture to include motion creation.
- The fallback mechanism effectively prevents unwanted motion and maintains the status quo to the greatest extent feasible, as evidenced by the demonstrations in the supplemental material.
- Motion tracking and motion transition were among the tasks the authors used the proposal for. Even when the fallback mechanism is activated, the suggested prediction approach can interpolate between motions in both tasks.
- A natural fallback mechanism (the ability to recognize impractical target motions and revert to reasonable behaviors found in the training set) is provided by the suggested method.
- The suggested paradigm generates more compact trajectory representations and expressive motions.

## Weaknesses

- Even with the actual latent space comparison, there appears to be a lack of reconstruction accuracy. As the prediction segment horizon N grows, the reconstruction accuracy decreases.
- More quantitative discussion of the fallback mechanism's efficacy is required. There's no hard proof that the mechanism performed well. If the motion prediction did not work when the fallback mechanism was in place, some statistics ought to be available.
- Restricted Ability to Adjust to Other Domains: The paper addresses robotics learning and motion representation. It might be more difficult to apply in other fields, like human motion analysis.

**Justification For Why Not Higher Score:**

As mentioned before, the ability to work on other domains is not proof.

**Justification For Why Not Lower Score:**

In conclusion, the paper offers a fresh and unique method for using GLD for motion representation and learning. It exhibits its excellence, lucidity, and importance in the field. The paper is a valuable contribution to the research community because of its innovative integration of preexisting ideas, its application to a new domain, and its resolution of previous results' limitations.

---

### Decision · Program_Chairs · 2024-01-16

Accept (spotlight)